

# Assimilating the dynamic spatial gradient of a bottom-up carbon flux estimation as a unique observation in COLA (v2.0)

Zhiqiang Liu[1,2], Ning Zeng[3,4,1], Yun Liu[5], Eugenia Kalnay[3], Ghassem Asrar[6], Qixiang Cai[1], Pengfei Han[7,1]

[1]Laboratory of Numerical Modeling for Atmospheric Sciences & Geophysical Fluid Dynamics, Institute of Atmospheric Physics, Chinese Academy of Sciences, Beijing, China
[2]University of Chinese Academy of Sciences, Beijing, China
[3]Dept. of Atmospheric and Oceanic Science, University of Maryland, College Park, Maryland, USA
[4]Earth System Science Interdisciplinary Center, College Park, Maryland, USA
[5]Dept. of Oceanography, Texas A & M University, College Station, TX, USA
[6]Universities Space Research Association, Columbia, Maryland, USA
[7]Carbon Neutrality Research Center, Institute of Atmospheric Physics, Chinese Academy of Sciences, Beijing, China

*Corresponding authors:* Zhiqiang Liu (liuzhiqiang@mail.iap.ac.cn) and Yun Liu (liu6@tamu.edu)

**Abstract.** Atmospheric inversion of high spatiotemporal surface $CO_2$ flux without dynamic constraints and sufficient observations is an ill-posed problem, and a priori flux from a "bottom-up" estimation is commonly used in "top-down" inversion systems for regularization purposes. Ensemble Kalman filter-based inversion algorithms usually weigh a priori flux to the background or directly replace the background with the a priori flux. However, the "bottom-up" flux estimations, especially the simulated terrestrial-atmosphere $CO_2$ exchange, are usually systematically biased at different spatiotemporal scales because of the deficiencies in understanding of some underlying processes. Here, we introduced a novel regularization algorithm into the Carbon in Ocean–Land–Atmosphere (COLA) data assimilation system, which assimilates a priori information as a unique observation (AAPO). The a priori information is not limited to "bottom-up" flux estimation. With the comprehensive assimilation regularization approach, COLA can apply the spatial gradient of the "bottom-up" flux estimation as a priori information to reduce the bias impact and enhance the dynamic information concerning the a priori "bottom-up" flux estimation. Benefiting from the enhanced signal-to-noise ratio in the spatial gradient, the global, regional, and grided flux estimations using the AAPO algorithm are significantly better than those obtained by the traditional regularization approach, especially over highly uncertain tropical regions in the context of observing simulation system experiments (OSSEs). We suggest that the AAPO algorithm can be applied to other greenhouse gas (e.g., $CH_4$, $NO_2$) and pollutant data assimilation studies.

## 1 Introduction

Climate change forced by the increasing atmospheric carbon dioxide ($CO_2$) concentrations threatens the health of the environment, living systems, and the economy. Therefore, it is essential to accurately estimate earth surface carbon fluxes (SCFs) and their variations for both scientific purposes and policy-making, including supporting the Paris Agreement (Byrne et al., 2022; Chevallier, 2021; Friedlingstein et al., 2022; Jiang et al., 2022; Weir et al., 2022; Deng et al., 2022). The SCFs





can be inferred from atmospheric $CO_2$ measurements using "top-down" (hereafter quotation marks will be omitted) techniques

of the Bayesian synthesis (e.g., Rodenbeck et al., 2003; Zammit-Mangion et al., 2022; Cho et al., 2022) and data assimilation (DA) techniques (e.g., Peters et al., 2007; Feng et al., 2009; Chevallier et al., 2010; J. Liu et al., 2014; Z. Liu et al., 2022). However, the top-down estimation could be ill-posed because of the sparseness feature of atmospheric $CO_2$ observations and systematic errors of the transport model and satellite retrieval (Basu et al., 2018; O'Dell et al., 2018; Yu et al., 2018; Schuh et al., 2019). To regularize the ill-posed problem, a priori SCFs precalculated from "bottom-up" (hereafter quotation marks will

be omitted) terrestrial models and ocean biogeochemical models are commonly applied (Baker et al., 2006; Peters et al., 2007). However, there are significant differences and biases among bottom-up SCFs due to the systematic deficiencies in current bottom-up models, which could contaminate the top-down SCF estimation (Philip et al., 2019; Fu et al., 2021).

Kang et al. (2011, 2012) developed a carbon data assimilation system by coupling an atmospheric general circulation model

with a local ensemble transform Kalman filter (LETKF) (Hunt et al., 2007). This system accurately estimated SCF at model grid resolution for the first time without directly applying a priori information by assimilating meteorology observations and $CO_2$ concentration observations simultaneously with a short assimilation window in observing system simulation experiments (OSSEs). Following this track, we developed the Carbon in Ocean–Land–Atmosphere (COLA) system (Liu et al., 2019; Liu et al., 2022), which maintains the LETKF with a short assimilation window and long observation window setting but replaces

the general circulation model with an atmospheric transport model, GEOS-Chem (Nassar et al., 2013). The COLA system also applies a constrained ensemble Kalman filter (CEnKF) technique to retain global mass conservation (Liu et al., 2022). As a result, COLA can provide good estimates of SCFs at model grid resolution along with atmospheric $CO_2$ concentration analyses.

On the other hand, even though a priori information includes biases, it could be used to further improve the SCF estimation in

COLA because it includes important dynamic information generated by terrestrial models, which is missing in the top-down inversion system. Therefore, it is worth exploring an appropriate a priori regularization method. The purpose of this paper is to introduce the development of the COLA (v2.0) system with a novel a priori regularization for SCF estimation, where a priori information is treated as a unique observation to be assimilated. This new approach is inspired by the CEnKF applied in COLA (Liu et al., 2022). With the comprehensive assimilation regularization, we are able to apply the spatial gradient of bottom-up

SCF estimations as a priori information to improve SCF estimation. The paper is organized as follows: section 2 introduces the COLA system and presents the new approach of a priori regularization for SCF estimation; section 3 compares a priori flux and its spatial gradient in the context of the signal/noise ratio and demonstrates and validates the new approach in the context of OSSEs; and section 4 provides the conclusion and discussion.

## 2   Data and Methods

### 2.1  Carbon in Ocean–Land–Atmosphere (COLA) DA system





The COLA DA system consists of an atmosphere transport model of GEOS-Chem, a local ensemble transform Kalman filter (LETKF) algorithm, a constrained EnKF, and the assimilated observations. Currently, we are using GEOS-Chem of version 13.0.2 driven by the Modern-Era Retrospective analysis for Research and Applications Version 2 (MERRA-2) meteorology reanalysis (Gelaro et al., 2017; The International GEOS-Chem User Community, 2021). GEOS-Chem requires the SCFs as

boundary forcings to simulate the atmospheric $CO_2$ concentration. The SCFs are usually generated from bottom-up SCF estimations, including terrestrial carbon fluxes ($F_{TA}$), terrestrial fire fluxes ($F_{IR}$), air-sea carbon fluxes ($F_{OA}$), and anthropogenic fossil fuel emissions ($F_{FE}$).

The DA algorithm used in the COLA system is LETKF, which was created by Hunt et al. (2007). LETKF is a powerful,

efficient deterministic EnKF variation. It is widely used for DA, including several operational centers, and was first used for carbon data assimilations by (Kang et al., 2011, 2012). LETKF estimates SCF as evolving parameters by augmenting it with the state vector $CO_2$. Similar to the other EnKF, the LETKF prefers a short assimilation window to produce accurate model state analysis, which reduces noise within the background for parameter estimation. On the other hand, parameter estimation requires a long training period to enhance the model response to the estimated parameter (the signal). Therefore, COLA

implements a new version of LETKF with a unique feature of a short assimilation window (1 day) and a long observation window (7 days) to enhance the SCF estimation (Liu et al., 2019).

Another unique feature of COLA is that it applies a constrained ensemble Kalman filter (CEnKF) to retain the global mass conservation of the system (Liu et al., 2022). The standard LETKF step updates the model state based on statistical information,

which could lead to model dynamic imbalance and loss of mass (Pan and Wood, 2006; Zeng et al., 2017; Janjić and Zeng, 2021). With CEnKF, an additional assimilation step is applied to the first guess of the global $CO_2$ mass as the observation with zero uncertainty to adjust $CO_2$ at each model grid point, ensuring the consistency between the analysis and the first guess after the standard LETKF procedure.

## 2.2 Assimilating a priori information as unique observations (AAPO)

Conceptually, COLA can further improve the SCF estimation with a priori regularization using the a priori generated from an independent bottom-up estimation. COLA treats SCFs as stationary parameters, where the SCFs are only updated by the LETKF statistically. There is no dynamic constraint for SCF estimation, and any gap in data could be filled with a priori information from a bottom-up estimation that includes the important dynamic information about the parameter of interest. There are two widely used a priori regularization approaches for EnKF-based carbon inversion systems. Feng et al. (2009)

treated the a priori information as a background for EnKF calculation,

$$\bar{f}_t^b = f_t^p, \tag{1}$$





where $f$ is the SCF; the bar denotes the ensemble mean; and the subscripts b, p, and t represent the background, a priori information of bottom-up estimation, and time, respectively. This approach omits useful information on the temporal dependency of SCFs, which is very important for resolving the subseasonal variation in SCFs, for example, in the COLA

system with a short assimilation window of 1 day. Peters et al. (2007) treated the a priori information as anchoring values and averaged it with SCF analysis as,

$$\bar{f}_t^b = \alpha \cdot \bar{f}_{t-2}^a + \alpha \cdot \bar{f}_{t-1}^a + (1-2\alpha) \cdot f_t^p, \qquad (2)$$

where $\alpha$ denotes the average weight of the analysis ranging from 0 to 1, and the subscript a represents the analysis. The averaging coefficient of $\alpha$ is ad hoc because it lacks a statistical foundation.


Here, we propose a new a priori regularization method to better follow the DA principle. The new approach treats the a priori information as unique observations to be assimilated into the COLA system in the LETKF analysis step (Fig. 1). Thus, the data include the atmospheric $CO_2$ observation and the a priori information,

$$y^o = [y_c^o, y_{ap}^o], \qquad (3)$$

where $y^o$ is the observation; the subscripts c and ap denote the $CO_2$ observation and the a priori information, respectively. The new regularization algorithm follows a similar perspective as Peters et al. (2007) but with a comprehensive EnKF approach. The a priori information used for regularization is not limited to SCF estimations. EnKF updates the parameter using the covariance between the parameter and observation variables. Therefore, a priori information can be any estimation related to bottom-up SCFs. As discussed before, a bottom-up estimation of SCFs includes important dynamic information as well as

significant biases. It will induce signal as well as noise when assimilated into the system. The impact of the signal must be larger than the impact of noise to obtain improved SCF estimation. Therefore, the estimation will benefit from the enhancement of the signal/noise ratio within the a priori information.

In COLA, the main purpose of applying a priori regularization is to introduce the dynamic constraint for SCF estimation. The

spatial gradient of a bottom-up estimation of SCF could be a better choice than the SCF estimation itself as the a priori information. We also speculate that the signal-to-noise ratio within the spatial gradient is larger than the SCF estimation itself. For example, the SCFs in a terrestrial model can systematically drift in a regionally spatially coherent manner because they are affected by the same model dynamic/physiological deficiency with a similar terrestrial eco-environment. The spatial gradient calculation can reduce the systematic drift. On the other hand, the SCF spatial gradient reflects the dynamic response of the terrestrial and oceanic systems to large-scale climate forcing, which is the information we seek. The signal-to-noise ratio

of SCF compared to its spatial gradient will be discussed in section 3.2.

Numerically, we define the SCF spatial gradient at a given grid as the SCF difference with its surrounding grids divided by the distance between them. Fig. 1 illustrates the assimilation steps of the COLA system. The atmospheric $CO_2$ observations and





the a priori SCF spatial gradient are assimilated at different LETKF steps to separate their impact, although both are treated as observations. In addition to the large-scale weather/climate forcing, the spatial gradient of the bottom-up SCF is also dependent on the local terrain, vegetation type, population, and urbanization. Therefore, the localization radius for the a priori SCF spatial gradient is set to a small value, and the SCF spatial gradient only constrains the local SCF.

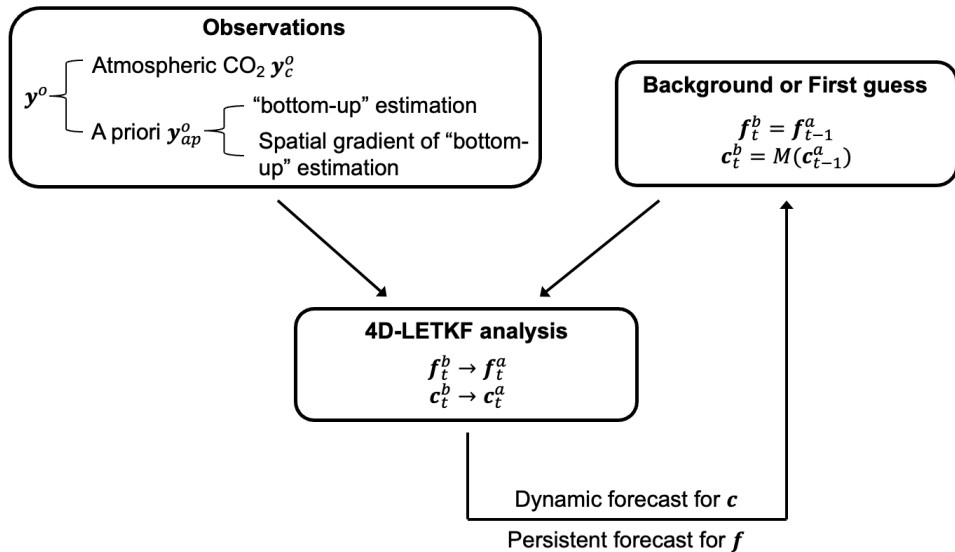

**Figure 1: The assimilation cycle of the COLA system, illustrating how and where the a priori is assimilated.**

### 2.3 SCF ensemble spread versus a priori uncertainty

COLA is a flow-dependent ensemble-based DA using the ensemble spread to represent the forecast and analysis uncertainties. The SCFs are temporally varying parameters to be estimated by the COLA. With a default setting, an estimated parameter is set as stationary during model integration (forecast step) and only updated by the DA algorithm in the analysis steps. As a result, the parameter ensemble spread remains unchanged during the forecast step and is reduced in the analysis step. This reduction leads to a progressively decreasing parameter ensemble spread, which eventually could result in a too-small parameter ensemble spread and filter divergence for parameter estimation. Furthermore, this parameter estimation procedure totally ignores the uncertainties related to the temporal variations in SCFs, thus further deteriorating the negative ensemble spread issue. Following Kang et al. (2012), we apply an additive inflation method to the SCFs to overcome this problem. This inflation method adds to each ensemble member with an anomalous SCF field chosen randomly from the bottom-up estimation of CASA within 30 days centered at the analysis time (Kang et al., 2012; Liu et al., 2019, 2022).

COLA assimilates the a priori SCF spatial gradients into the system, which needs to define the a priori uncertainty. In this study, we simply set the a priori uncertainty proportional to the uncertainty of the analysis ensemble uncertainty,










$$\sigma_{i,t}^{\mathrm{ap}} = \alpha \cdot \sigma_{i,t}^{\mathrm{a}}, \tag{4}$$

where $\sigma_{i,t}^{\mathrm{ap}}$ is the a priori uncertainty of the bottom-up estimation or the spatial gradient of the bottom-up estimation, and $\alpha$ is the scaling factor. This approach is a simple adjustment for the a priori uncertainty. We find an optimal scaling factor of 5 based on several tests. In reality, a bottom-up SCF estimation product may come with its uncertainty estimation. We may derive

the uncertainty of the SCF spatial gradient from it. The importance and impact of those uncertainties and whether their accuracies are good enough for DA application remain to be further explored in the future.

## 3    Observing system simulation experiments (OSSEs)

### 3.1    Experimental setup

In this section, we evaluated the new algorithm in the context of OSSEs. The experimental period spans from 1 September

2014 to the end of 2015, with a spin-up period in 2014. GEOS-Chem is running with a horizontal resolution of 4°×5° and 47 hybrid pressure-sigma vertical levels for $CO_2$ simulation. We assimilate both the surface and satellite data as described in Liu et al. (2022). The surface data are obtained from the $CO_2$ GLOBALVIEWplus v8.0 ObsPack. For satellite data, we used a 10-second averaged ACOS v10 level two retrieval of land-nadir and land-glint from the Orbiting Carbon Observatory-2 (OCO-2) (Crisp et al., 2017; O'Dell et al., 2018; Baker et al., 2021; Cox et al., 2022). Following Liu et al. (2022), the observation

networks and their error scales are used to create the pseudo observations in the OSSEs. We set the $CO_2$ observation localization radius to 4000 kilometers.

A nature run is driven by the $F_{OA}$ from Rödenbeck et al. (2014), $F_{FE}$ from the Open-source Data Inventory of Anthropogenic CO2 emissions (ODIAC) (Oda et al., 2018), and the $F_{IR}$ and $F_{TA}$ generated from the terrestrial model of the VEgetation Global

Atmosphere Soils (VEGAS) model (Zeng et al., 2005), which serves as the "truth" for the OSSE. The $F_{TA}$ is chosen as the flux to be estimated. The other fluxes of $F_{IR}$, $F_{FE}$, and $F_{OA}$ are set to be identical in the nature run. Another terrestrial model of CASA (Potter & Klooster, 1997) is used to provide an independent bottom-up $F_{TA}$ estimation for the a priori regularization in our experiments. The difference between the $F_{TA}$ from VEGAS and CASA represents the bias among bottom-up estimations related to terrestrial model deficiencies.

### 3.2    Signal-to-noise ratio (SNR) analysis

In this section, we mimic the typical bias of a priori information and derive its signal-to-noise ratio (SNR) based on two different terrestrial model estimations of the VEGAS and CASA. The estimations of VEGAS and CASA are treated as the truth and the a priori information, respectively. The variations in the truth are regarded as the signal. The differences between the "truth" and the a priori information are regarded as the noise. We define the SNR of the spatial gradient and the $F_{TA}$ as,





$$\mathbf{SNR}^{SG}(x, t) = \frac{|\overrightarrow{\mathbf{f}^{TSG}(x,t)} - \overrightarrow{\mathbf{f}^{TSG}(x,t-1)}|}{|\overrightarrow{\mathbf{f}^{PSG}(x,t)} - \overrightarrow{\mathbf{f}^{TSG}(x,t)}|}, \tag{5}$$

$$\mathbf{SNR}^{S}(x, t) = \frac{|\mathbf{f}^{T}(x,t) - \mathbf{f}^{T}(x,t-1)|}{|\mathbf{f}^{P}(x,t) - \mathbf{f}^{T}(x,t)|}, \tag{6}$$

where the superscripts PSG and P denote the a priori of SCF spatial gradient and SCF, respectively; the superscripts TSG and T denote the truth of SCF spatial gradient and $F_{TA}$, respectively; and x denotes the grid point location at time t.

Both $\mathbf{SNR}^{SG}$ and $\mathbf{SNR}^{S}$ over the Northern Hemisphere are larger than those over the tropics and Southern Hemisphere (Fig. 2a, b), which is consistent with the larger $F_{TA}$ seasonal cycle over the Northern Hemisphere than over the tropics and Southern Hemisphere. The $\mathbf{SNR}^{SG}$ is significantly greater than $\mathbf{SNR}^{S}$ nearly everywhere (Fig. 2c), indicating that the spatial gradient is better than SCF itself in terms of SNR. The $\mathbf{SNR}^{SG}$ is almost double the corresponding $\mathbf{SNR}^{S}$ in the Sahel, southern Africa, tropical South America, and eastern Siberia, where the noise of the a priori $F_{TA}$ is significantly larger than that in the other areas (Fig. 4a).

Here, we use VEGAS and CASA estimations as examples to reveal that the spatial gradient has a larger SNR than the $F_{TA}$ itself. The same comparison can be made based on other terrestrial model estimations. Conceptually, DA induces both signal and noise into a system by assimilating observations with errors. A DA application expects the impact of the signal to dominate and offset the impact of noise. Therefore, it tends to achieve better analysis by increasing the SNR of assimilated observations. Thus, we speculate that the spatial gradients are better than the flux itself as a priori information. Since the current understanding of the carbon cycle processes is highly uncertain in the tropics and Eurasia (O'Sullivan et al., 2022), the two models we chose differ significantly in these areas, which is a good proxy for the real-world scenario.

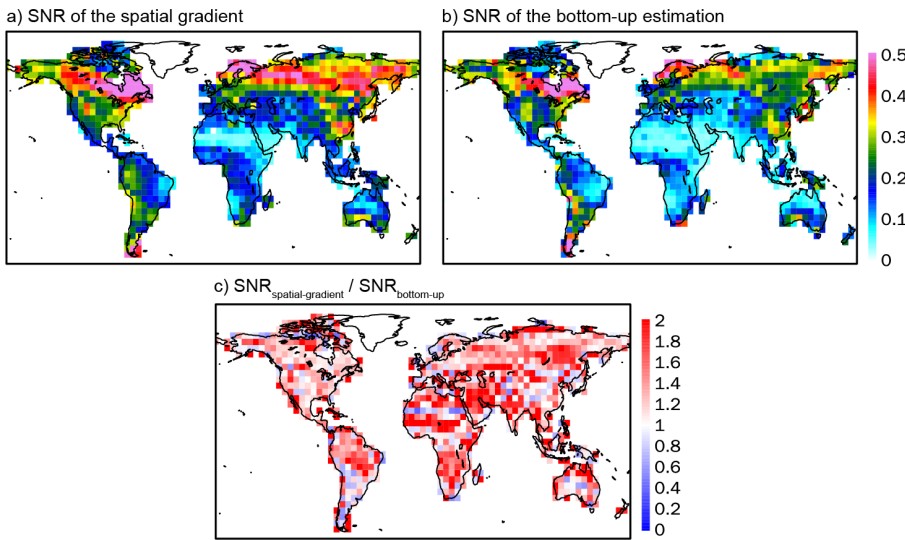

**Figure 2: a) The annual mean signal-to-noise ratio pattern of the a priori $F_{TA}$ spatial gradient. b) The same as a) but**





for the a priori $F_{TA}$. c) Annual mean signal-to-noise ratio of the spatial gradient divided by the signal-to-noise ratio of the $F_{TA}$.

### 3.3 OSSE results

Four experiments are performed with COLA in the context of OSSE to evaluate the proposed a priori regularization approach. The first experiment estimates the $F_{TA}$ by only assimilating the synthetic observations of the atmospheric $CO_2$ concentration (EXP-NP). Based on the first experiment, experiment two (EXP-P) uses the approach described in Eq. (2) with a priori $F_{TA}$ from CASA and $\alpha$ of 0.2; experiment three (EXP-ASG) applies the proposed assimilation approach with a priori $F_{TA}$ spatial gradient in the CASA estimation; experiment four (EXP-AP) assimilates the a priori $F_{TA}$ of the CASA estimation. We compare

EXP-ASG with the other three experiments to illustrate the improvement of SCF estimation. The evaluation metrics of root-mean-square error (RMSE) and mean bias are defined as,

$$\mathbf{RMSE}^A(x) = \sqrt{\frac{\sum_{t=t1}^{t2}(f^A(x,t)-f^T(x,t))^2}{t2-t1}}, \tag{7}$$

$$\mathbf{RMSE}^{ASG}(x) = \sqrt{\frac{\sum_{t=t1}^{t2}(|\vec{f}^{ASG}(x,t)-\vec{f}^{TSG}(x,t-1)|)^2}{t2-t1}}, \tag{8}$$

$$\mathbf{MB}^A(i) = \frac{\sum_{t=t1}^{t2}(f^A(i,t)-f^T(i,t))}{t2-t1}, \tag{9}$$

where x denotes a grid point; the superscripts A and T denote the analysis of $F_{TA}$ and truth, respectively; ASG denotes the analysis of the spatial gradient of $F_{TA}$; all are calculated for the time from t1 and t2.

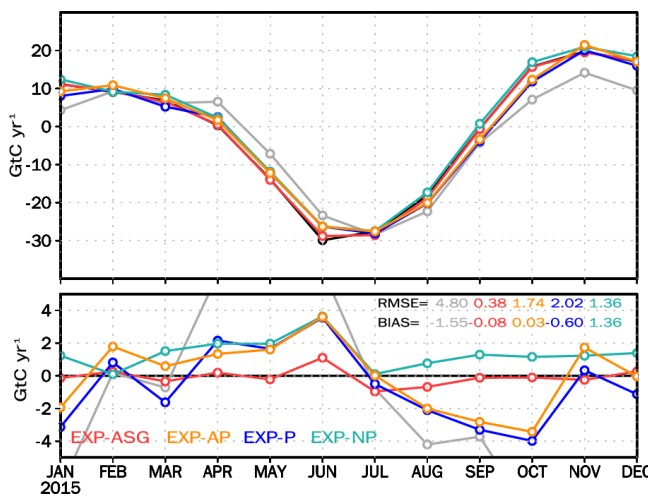

**Figure 3: The top figure is the global total seasonal cycle of the truth (black), the a priori $F_{TA}$ (gray), and the four**
**assimilation experiments. The bottom figure is the global total difference of the four assimilation experiments compared to the truth. The annual mean RMSE and mean biases of the a priori $F_{TA}$ and the four experiments are denoted at the upper right corner of the bottom figure.**

First, we focus on the global total $F_{TA}$ (Fig. 3). The estimated errors are shown in annual mean flux biases and the RMSE of
the seasonal cycle. The flux biases of EXP-P and EXP-AP are -0.60 GtC yr⁻¹ and 0.03 GtC yr⁻¹, respectively, which are



significantly smaller than those of EXP-NP (1.36 GtC yr$^{-1}$). The improvement is mainly due to the negative bias of $F_{TA}$ from CASA compromising the positive bias of the estimates in EXP-NP without a priori regularization. The EXP-ASG removes most of this bias with only 0.08 GtC yr$^{-1}$ remaining.

The RMSE of $F_{TA}$ in EXP-ASG is significantly smaller than those with other experiments. Instead of improving the estimation, the a priori regularization with the $F_{TA}$ from CASA degrades the seasonal cycle estimation of the global $F_{TA}$. Both EXP-P and EXP-AP show an increase in RMSE of $F_{TA}$ concerning EXP-NP because of the very large seasonal cycle errors within the a priori $F_{TA}$ used for regularization (Fig. 3). The new AAPO regularization is clearly better than the traditional method even with the same biases for the a priori $F_{TA}$ from CASA. The bias and RMSE for EXP-AP are reduced by 66% and 24% compared to

EXP-P, respectively. The AAPO regularization with an improved a priori $F_{TA}$ spatial gradient further improved the estimation. The bias and RMSE for EXP-ASG are reduced by 87% and 81% compared to EXP-P, respectively.

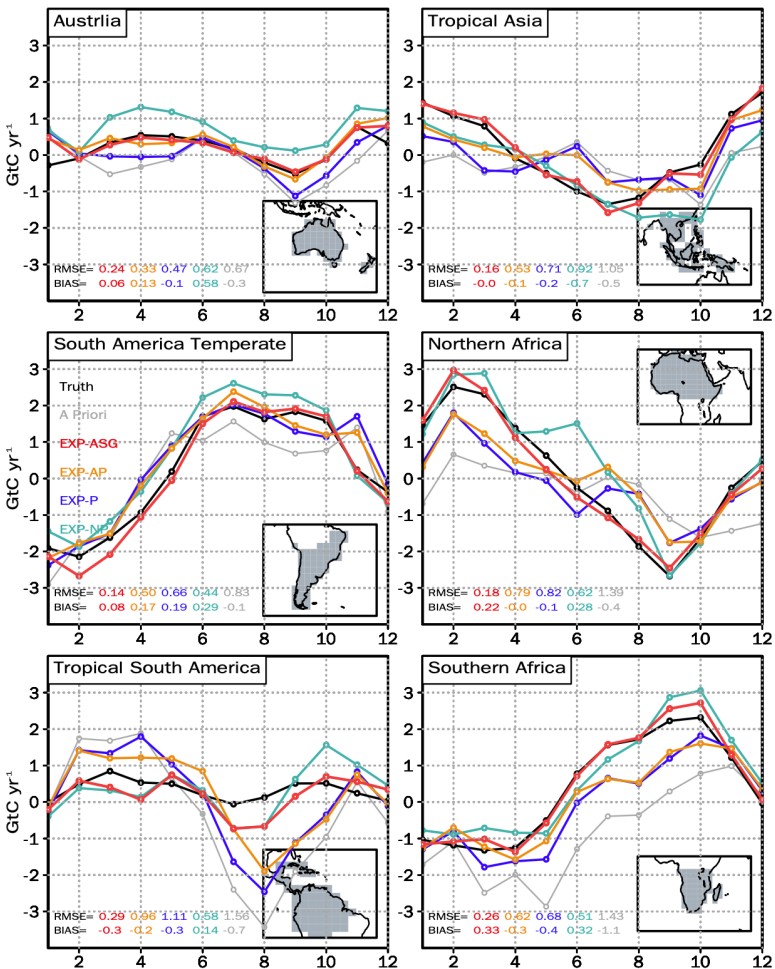

**Figure 4: The seasonal cycles of the regional $F_{TA}$ for truth (black), a priori (gray), EXP-ASG (red), EXP-AP (orange),**





**EXP-P (blue), and EXP-NP (green) in the tropics and Australia. The values in the bottom-left corner of all figures are the RMSE, the mean bias of the four experiments, and the a priori $F_{TA}$ compared to the truth $F_{TA}$. The regions are defined based on the OCO2MIP mask (Crowell et al., 2019).**

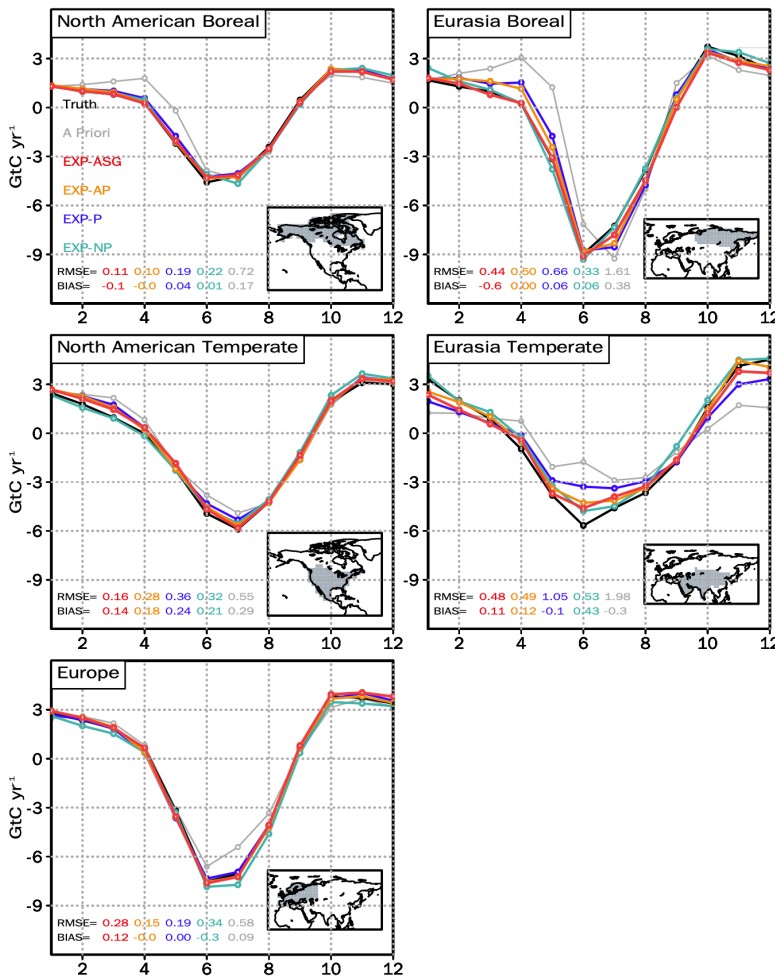

**Figure 5: Same as in Fig. 4, but for the Northern Hemisphere regions.**

At the regional scale, defined based on the OCO2MIP mask (Crowell et al., 2019), all four experiments reproduce the Northern Hemisphere seasonal cycle accurately, which is mainly because of the better observation coverage and less biased a priori $F_{TA}$ as compared to the Southern Hemisphere regions (Fig. 5). In the Southern Hemisphere, significant differences exist between

the "truth" $F_{TA}$ and the a priori $F_{TA}$ from CASA. The differences are larger than the RMSE of the estimated $F_{TA}$ of EXP-NP for most regions, especially South America Temperate, Tropic South America, Northern Africa, and Southern Africa, where a priori $F_{TA}$ degrades the $F_{TA}$ estimation with the RMSEs increasing for EXP-P and EXP-AP compared to EXP-NP. The seasonal estimation of EXP-ASG is significantly better than the other experiments in terms of RMSE (Fig. 4). The regional seasonal estimation of EXP-AP is better than that of EXP-P, which indicates that the AAPO regularization is better than the traditional

method, and this is consistent with the global seasonal results presented earlier. In tropical South America, the seasonal amplitude of the a priori $F_{TA}$ is significantly larger than the truth of $F_{TA}$, which greatly contaminates the estimation in EXP-AP and EXP-P. A similar phenomenon also appears in some other regions (e.g., northern and southern Africa) where the a priori and the truth of $F_{TA}$ show a large discrepancy. However, the a priori $F_{TA}$ spatial gradient can help separate the biased information of the $F_{TA}$ and significantly improve the seasonal estimation in regions where the a priori $F_{TA}$ is not good and

observations are rare.

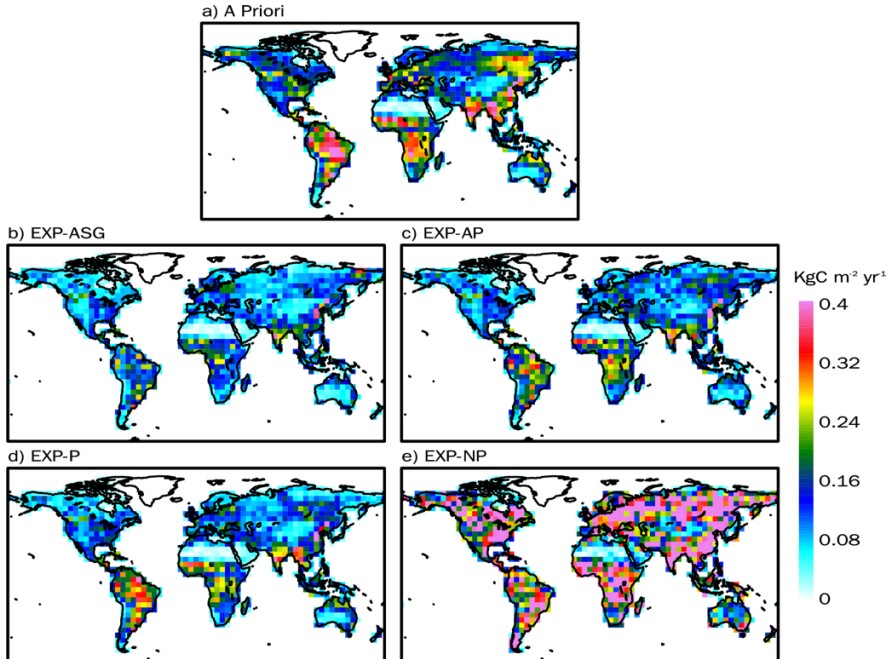

**Figure 6: The RMSE of the $F_{TA}$ in (a) the a priori, (b) EXP-ASG, (c) EXP-AP, (d) EXP-P, and (e) EXP-NP calculated based on Equation 7 from January to December 2015.**


At the grid-point scale, EXP-NP produces a very noisy $F_{TA}$ with abnormally large RMSEs of the estimated $F_{TA}$ because LETKF overfits the gridded $F_{TA}$ with limited observations (Fig. 6e). Therefore, a priori regularization is expected to enhance the results for COLA. The RMSEs in EXP-P, EXP-AP, and EXP-ASG are significantly reduced compared to EXP-NP. Large errors exist in the a priori $F_{TA}$ over the South American, African, and Indian regions (Fig. 6a). When the a priori $F_{TA}$ is used for

regularization, those errors are also introduced into the system resulting in the final $F_{TA}$ estimation with a large RMSE shown in the same locations, especially for EXP-P (Fig. 6b, c). Those errors are less severe within EXP-AP with the AAPO regularization applied concerning EXP-P. The use of AAPO with an improved a priori $F_{TA}$ spatial gradient avoids those large errors within the a priori $F_{TA}$ and produces much better estimation in EXP-ASG. At the model grid scale, EXP-ASG significantly outperformed EXP-AP and EXP-P for most grid points.






In addition to the RMSE of $F_{TA}$, we also analyzed the RMSE of the $F_{TA}$ spatial gradient. For EXP-AP and EXP-P, which assimilate or add the a priori $F_{TA}$, the spatial gradient information is digested simultaneously within the original $F_{TA}$. Similar to the RMSE of $F_{TA}$, EXP-NP also produced very noisy spatial gradient estimation (Fig. 7e). With the help of the two types of a priori information, the other three experiments significantly reduced the RMSE concerning EXP-NP (Fig. 7b-d). However,

for EXP-ASG, which directly assimilates the spatial gradient information, its RMSE is smaller than that of EXP-AP and EXP-P. Moreover, the RMSEs of EXP-AP and EXP-P are almost identical to the RMSE of the a priori $F_{TA}$ spatial gradient at most grid points, indicating that the second-order spatial gradient information is not effectively digested in the two experiments and that the first-order biases in the original a priori $F_{TA}$ dominate the resulting information.

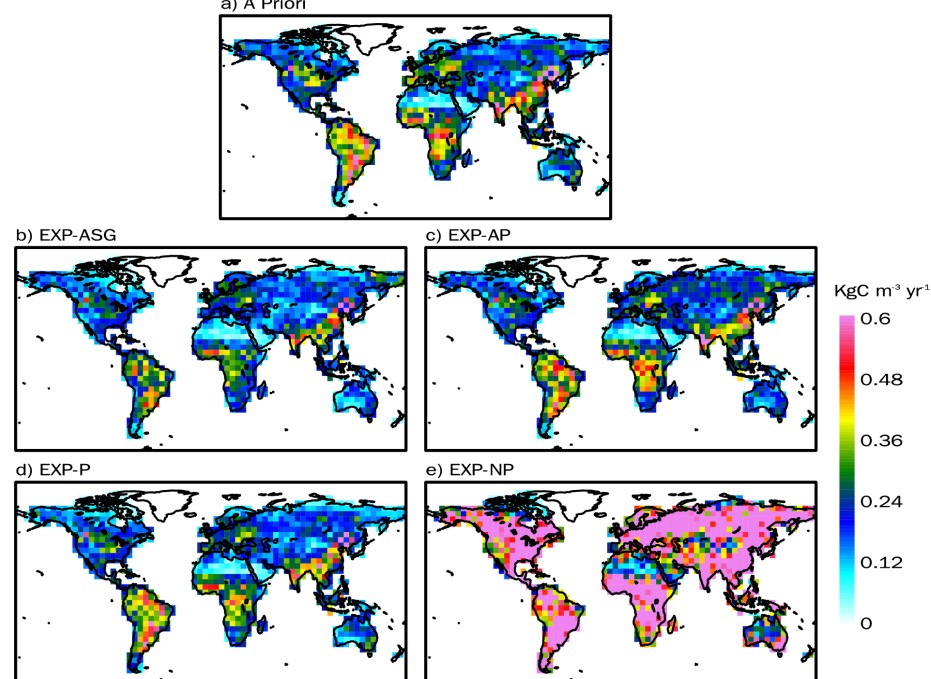


**Figure 7: The RMSE of the spatial gradient of $F_{TA}$ in (a) the a priori, (b) EXP-ASG, (c) EXP-AP, (d) EXP-P, and (e) EXP-NP calculated based on Equation 8 from January to December 2015.**

## 4    Summary and Discussion

In this study, we developed a novel algorithm for the ensemble-based $CO_2$ inversion system, in which the spatial gradient of a
bottom-up model estimation is dynamically assimilated as a unique observation (AAPO). The AAPO algorithm that uses the spatial gradient of a bottom-up model estimation as the a priori information aims at separating the first-order systematic biases in the bottom-up model estimations out of the inversion framework from a comprehensive DA perspective. In the context of OSSEs that assimilate in-situ and OCO-2 land-nadir and land-glint observations, it significantly overperformed the traditional



scheme that directly adds the a priori flux to the first guess. As a result, the spatial gradient consideration helps improve the

accuracy of regional flux estimation, especially over the highly uncertain tropics.

EnKF is widely used in state data assimilation studies, such as weather forecasting, for its flexibility, efficiency, and error transport feature. However, the advantage of error transport is partly sacrificed or abandoned by introducing the a priori flux information to the background in most of the EnKF-based $CO_2$ inversion methods (Peters et al., 2007; Feng et al., 2009). This

is because of the loss of a dynamic model to provide the background and the background covariance estimations. Different from most EnKF-based systems, COLA maintains the mean and error transport advantages of the EnKF by including the dynamic information constraints of the a priori flux spatial gradient and using an additive covariance inflation method (Liu et al., 2022).

In addition to the $CO_2$ inversion problem investigated in this study, the proposed new assimilation algorithm can also be applied to ensemble-based source/sink inversion studies of methane, nitrogen dioxide, and other chemical species. Based on the concept of applying constraints as unique observations, the spatial gradient constraint can be introduced to variational methods. With the increasing number and spatial coverage of $CO_2$ observations and the improving accuracy of atmospheric transport and satellite retrieval algorithms, the dependency on the a priori estimation is expected to be further reduced. The proposed

new approach in this study offers a unique strategy and a new approach for improving the estimation of geophysical parameters and greenhouse gas fluxes, especially for observation- and understanding-limited regions of the world.

*Code and data availability.* The code for the new AAPO regularization scheme can be accessed at https://doi.org/10.5281/zenodo.7592827.

*Author contributions.* ZL and YL conceived the a priori regularization scheme. ZL, YL, and NZ designed the OSSEs. ZL wrote

the code and ran the OSSEs. ZL, NZ, YL, and EK developed the system. QC supplied the VEGAS model output. ZL, YL, NZ, and GA wrote the paper. All authors contributed to the preparation of this paper.

*Acknowledgments.* We thank the OCO-2 science team and the NOAA greenhouse gas team for providing the $CO_2$ observations. Special thanks to Dr. David Baker for processing the 10-second averaged OCO-2 retrievals and Drs. Sourish Basu and Tom Oda for providing the hourly ODIAC data. We are also grateful to the OCO-2 Flux MIP colleagues for their valuable feedback

and discussion of COLA and CEnKF.

*Financial support.* This work was supported by the National Key R&D Program of China (No. 2017YFB0504000).



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
