# Peer review of "Assimilating the dynamic spatial gradient of a bottom-up carbon flux estimation as a unique observation in COLA (v2.0)"

_Geoscientific Model Development, 2023_

## Referee Comment (RC2)

**Referee comments on manuscript  gmd-2023-15**
Anonymous:  28 April 2023

In this manuscript, an alternative *a priori* flux constraint is presented in the context of a global $CO_2$ flux inversion performed using an ensemble Kalman filter (EnKF) with a short assimilation window.  Observing system simulation studies (OSSEs) are preformed to give an idea of how this alternative constraint might function when used with real data in a real inversion.  The flavor of EnKF used is the local ensemble transform Kalman filter (LETKF), as implemented in the Carbon in Ocean–Land–Atmosphere (COLA) data assimilation system, a global $CO_2$ flux inversion based on the GEOS-Chem transport model.

The alternative flux constraint is formulated in terms of the spatial gradient of the fluxes: finite differences of flux using adjacent grid boxes in the model.  These spatial gradients are then added as new measurements in the measurement vector, as opposed to additional constraints in the traditional *a priori* state vector.  Gradients used in this manner could capture the bulk of the flux constraint (its spatial and temporal patterns), while at the same time cutting the tie to the absolute value of the flux -- i.e. its overall constant offset or long-term mean.  This in turn could be useful when using priors for which the variability is more robust than the long-term mean -- for example, the terrestrial biosphere models used as priors for $CO_2$ fluxes over land in global flux inversions, which do a good job getting the seasonality of the fluxes right (e.g., using satellite measurements vegetation greenness, plus assumptions on the timing of respiration) but a less-good job of estimating the integrated flux across a full year.  By getting rid of the constraint to the long-term mean of the prior, the flux estimate might be freer to move to the long-term mean given by the data and not suffer from being biased in the direction of the incorrect or inaccurate prior.  This of course would be at the cost of losing any benefit that that long-term prior mean might provide.

In general, a flux constraint of this nature should be able to be implemented as a measurement in the measurement vector, as is done here, assuming that the measurement uncertainty used gives the constraint the same weight as it would have had if it had been implemented more traditionally in the *a priori* state vector.  One would have to avoid double counting by not also having the traditional flux prior in force at the same time.

In their OSSE experiments, the authors compare the effectiveness of this flux spatial gradient constraint against the usual prior flux constraint (i.e. in terms of

the actual flux value itself, not the spatial gradient) implemented either in the measurement vector or, more traditionally, as part of the *a priori* state vector; in the latter case, a couple different forms for the first guess of the flux at the new measurement time are used: either 1) a combination of the prior flux at the given time plus the flux estimate from the EnKF at the two immediately-earlier times, or 2) just the prior flux at the new time. This is done using one land biospheric model (VEGAS) to generate the 'true' measurements, and a second model (CASA) to be used as the prior flux. The authors find that, in general, when the flux gradient prior is used, the EnKF does a better job estimating the true fluxes than when three other approaches based on the absolute fluxes themselves (i.e., not gradients) are used.

While these results look promising, there are some inconsistencies in the results that I would like explained. Also, I suggest modified OSSEs in which the ocean fluxes are allowed to be corrected along with the land fluxes, in order to give a more realistic test of the new constraint. Finally, there is a lack of detail in the description of the methods used that makes it difficult for me as a reviewer to assess the full meaning of the results. I suspect that the general reader will have similar questions. I suggest that the authors add these needed details to the manuscript, address the points that I raise below, and resubmit, at which point I will re-review it and decide on final publication.

Comments:

First, the authors should describe in detail [with equations] the meaning of the terms 'assimilation window' and observation window', since how these terms are used in the context of the LETKF is not generally known. The reader should not have to go back to the previous LETKF papers to find this. Does the 1-day assimilation window mean that the filter is stepped forward in time a day at a time, each day allowing the new measurements to update the fluxes across the 7-day measurement window (i.e. the current day plus six previous days)? If so, the weight given to the flux constraint (or flux prior constraint) for each of those 7 days ought to be reduced, so that the integrated effect of the seven days of measurement updates affecting the fluxes on a given day is equivalent to the weight given to a single days' flux prior in some other estimation method (e.g. a variational method or a matrix-inversion-based Bayesian synthesis method).

Second, the weights given to the spatial gradient constraint in the inversion relative to the straight flux constraint cases ought to be given. Perhaps the spatial gradient case does a better job because it has a looser (or tighter)

weighting than the other cases.  A tighter flux prior usually results in a worse fit to the measurement data; or, vice versa, the inversion can over-fit the measurement data at the cost of too great a change from the flux prior.  Knowing the weights assumed in the inversion for the gradient case *vis a vis* the straight flux case could help assess this.  Similarly, some information on how good the fit to the measurement data is for the four cases could help.

Third, if the flux constraint can be implemented equally as well in the measurement vector as in the *a priori* state vector, then the two cases in which the straight flux prior are implemented these two ways should give the same flux results.  That is, the EXP-NP case, in which the flux prior is applied normally, as the *a priori* constraint on the fluxes in the state vector, and the EXP-AP case, in which the flux prior is assimilated as a measurement in the measurement vector, should give the same flux estimates.  But they don't -- they give quite different answers, as seen by the turquois and orange lines in Figures 3 through 5.  What is it about the different implementation of the prior that causes these differences?  Different weights used in each case?  A different number of times that the constraint is applied (if fluxes at multiple times are updated by measurements at a single time)?  Similarly in Figures 6 and 7, the EXP-NP case gives much worse RMSEs for flux and flux spatial gradient than does EXP-AP.  Why is this, if the two ways of implementing the prior are equivalent?  I can understand why, with a short-window inversion, the EXP-NP case might have higher values for these metrics (i.e. a flux error frozen in at a given assimilation step would need to be corrected by a balancing error at the next step of opposite sign, resulting in a lot of noise in time), but what is it about the EXP-AP implementation that prevents this?

Fourth, because the OSSE experiments use the same ocean fluxes in the truth and assimilation runs, there is effectively no error coming from the oceans and no need to allocate any flux corrections there in the inversions.  This is effectively the same thing as holding the oceans fixed and only allowing flux changes over the land areas.  This significantly simplifies the inversion and gives an overly-optimistic view of how well the inversions can retrieve the land fluxes.  However, even worse, it may favor the spatial gradient prior constraint more than the straight flux prior constraint, since, with the ocean corrections fixed to zero, the fluxes bordering the oceans are then strongly constrained by the spatial gradient constraint, and the fluxes in the interior similarly prevented from moving as much as they otherwise would.  With the straight flux constraint, however, the fluxes are still allowed to trade off corrections between continents.  It would be interesting to see whether these same favorable results with the EXP-ASG case

are achieved if more realistic errors are allowed over the oceans (i.e., if separate ocean flux models were used in generating the truth and prior, as has been done with the land biospheric fluxes here).

Fifth, it would be useful for the authors to discuss how specific their results are to the flux inversion method they use (a short-window EnKF). Would they anticipate that the alternative flux spatial gradient constraint would give similar improvements in methods that allow the transport model to link measurements and flux corrections across a longer span? Similarly, since this reliance on the transport model is less important when there is more data coverage, would the results obtained here still hold were a less-dense observing network (the *in situ* $CO_2$ network instead of a $CO_2$-measuring satellite, say) to be used?

More-detailed comments:

14: "dynamic constraints" I do not believe that the reason the inversion problem is ill-posed is because of the lack of explicit dynamical constraints in the setup. Really it is due to the sparse data.

16-17: "Ensemble Kalman filter-based inversion algorithms usually weigh a priori flux to the background or directly replace the background with the a priori flux." It is not very clear what this means. Please reword. What do you mean by 'background'?

21: spell out "AAPO"? It is not clear why you use this combination of letters for what you are describing.

38: I wouldn't say the problem is 'ill-posed' because of transport errors or retrieval biases -- those just bias the result. Ill-posedness is more due to lack of a sufficient data constraint, for example, trying to solve for more unknowns than can be constrained by a given number of data points.

49: "the LETKF with a short assimilation window and long observation window setting"
I do not see this described later in the text. Please describe what these 'window' terms refer to, for example in terms of the filter time stepping, what span of data is assimilated at each time step, and what span of fluxes is allowed to change per time step; preferably with equations.

54-56: "On the other hand, even though *a priori* information includes biases, it could be used to further improve the SCF estimation in COLA because it includes important dynamic information generated by terrestrial models, which is missing in the top-down inversion system."
It is not clear why you think that dynamic information generated by the terrestrial models is not represented in the top-down inversion systems. Insofar as it is used to generate the *a priori* SCFs, it is in there. Do you mean to say that the dynamical constraint of the *a priori* fluxes is not represented explicitly as a dynamic model in the Kalman filter, i.e. as a formal constraint?

75: add "at" after "including"

77-81 "Similar to the other EnKF, the LETKF prefers a short assimilation window to produce accurate model state analysis, which reduces noise within the background for parameter estimation. On the other hand, parameter estimation requires a long training period to enhance the model response to the estimated parameter (the signal). Therefore, COLA implements a new version of LETKF with a unique feature of a short assimilation window (1 day) and a long observation window (7 days) to enhance the SCF estimation (Liu et al., 2019)."

It is not clear how these various 'windows' relate to the fluxes being solved for. You should write out with equations what is being solved for, how the time stepping is done, what observations are assimilated in which time step with which weights, etc. And point out which spans are the 'observation window' versus the 'assimilation window'. This may be detailed in previous LETKF papers, but the reader shouldn't have to go back to them to understand what is being used here.

119: "In COLA, the main purpose of applying *a priori* regularization is to introduce the dynamic constraint for SCF estimation."

It is not at all clear that you have now introduced a better dynamic constraint by changing from using the prior flux value to using spatial gradients instead. Nothing involving dynamics has been changed by this. All you have succeeded in doing is removing the link to the overall absolute value of the prior flux (the long-term mean). That may indeed have value, but don't confuse it with dynamics. Any dynamics that were or were not in the original flux prior are still there with this new constraint. Please reword to reflect this, here and elsewhere in the document where 'dynamics' are discussed.

138-147: You are free to add dynamical noise to your propagation of information

forward in time in your model.  You should discuss why you choose not to add dynamical noise that reflects errors in your transport model and/or variability in the land fluxes not captured by a forward propagation based on persistence. Why do you instead add an inflation term that is based more on the technical needs of your EnKF rather than a physically-based dynamical error?

149-150: "COLA assimilates the *a priori* SCF spatial gradients into the system, which needs to define the *a priori* uncertainty. In this study, we simply set the *a priori* uncertainty proportional to the uncertainty of the analysis ensemble uncertainty."   Please describe what this analysis ensemble uncertainty looks like. Does it differentiate between forested areas that are likely to have larger fluxes and flux uncertainty and desert areas that are likely to have smaller ones?  (Or similarly for flux gradients?)  A sensitivity study done using uncertainties proportional to the magnitude of the fluxes in either the VEGAS or CASA models, or based on the difference between VEGAS and CASA (and preferably other models), would be welcome to test the dependence of your results on this assumption.

165-166: "We set the CO2 observation localization radius to 4000 kilometers." Since the general reader probably will not understand what this means, please say what this means, practically, in your inversion setup.  Does it mean literally that each observation has zero impact on any flux farther away than 4000 kilometers at a given time?  What about at previous times?

168-174:  By using the same fossil fuel, ocean, and wildfire fluxes in both the truth and
prior, the simulation is artificially rosy:  terrestrial fluxes are solved for
using only differences there by permitting flux corrections only over the land and not over the ocean.   By not considering the impact of ocean flux errors, this will give you
lower error estimates for the land fluxes than you'd get otherwise.  It would be a useful sensitivity study to look at the impact of considering ocean flux errors, as well.
Figures 6 & 7:  The difference between the EXP-NP and EXP-AP cases still needs to be explained.  Yes, the short window of the COLA setup results in over-fitting of the data and noisy fluxes (and spatial gradients) in the EXP-NP case.  But how does applying the prior flux constraint via the measurement vector prevent this?

290: What does 'dynamically' in 'dynamically assimilated' indicate?  Is this some special sort of assimilation method?   Also, define what the acronym 'AAPO' refers to.

297-304: "However, the advantage of error transport is partly sacrificed or abandoned by introducing the *a priori* flux information to the background in most of the EnKF-based $CO_2$ inversion methods (Peters et al., 2007; Feng et al., 2009). This is because of the loss of a dynamic model to provide the background and the background covariance estimations. Different from most EnKF-based systems, COLA maintains the mean and error transport advantages of the EnKF by including the dynamic information constraints of the a priori flux spatial gradient and using an additive covariance inflation method (Liu et al., 2022)."
I agree that the loss of the dynamical model for the fluxes in most of our flux inversion methodologies is unfortunate.  I do not believe, however, that you are remedying that with your spatial gradient constraint here.  Nothing has changed regarding the dynamics in using this constraint.  Your only change is to cut the tie to the long-term mean, allowing your estimate to be shifted up or down as a whole more easily.

310: 'unique strategy'?  Maybe referring to it as a 'new strategy' would be better.

---

## Author Comment (AC1)

**Reply to Reviewer 1:**

**General comments**

This paper presents a new algorithm for ingesting information about surface CO2 fluxes from bottom-up estimations into a top-down atmospheric inversion system. The main innovation is that instead of using the bottom-up fluxes as a priori fluxes in the inversion, the spatial gradients of the fluxes are instead assimilated as observations, which the authors argue increases the signal-to-noise ratio and avoids contaminating the analysis with biases in the bottom-up fluxes.

While the idea of assimilating the spatial flux gradients rather than the fluxes themselves is interesting and worth investigating, I had a hard time assessing the manuscript on scientific quality and validity because of incomplete information, confusing use of terms, and distracting issues in the text and figures. More detailed comments are provided below. I believe addressing these issues goes beyond the scope of a major revision and recommend the authors to substantially rework the manuscript in terms of content and presentation. Only then can a proper assessment be made about the scientific quality.

Reply: Many thanks for your constructive comments and suggestions. We acknowledge that we have missed some important information and the experiments are not optimal. We are sorry for the inconvenience. We have reworked the manuscript in terms of method, experiment setup, and results. And the manuscript was also polished by AJE. The main changes are as follows:

1.  Information: Adding more details about the short assimilation window and long observation window / experiment setup of initial condition, ensemble size etc. / observation localization / additive inflation / generation of pseudo observation.

2.  Clearer method description: a) Assimilating $CO_2$ observation before assimilating a priori (Fig. 1). b) Using $\nabla f$ to represent the spatial gradient.

3.  New experiment setup: a) OCO2+insitu vs insitu only. b) Making sure the weight of a priori is identical in different experiments. c) Changing CASA to SiB4 as the a priori.

4.  Clearer message from the OSSE results: Better hemispheric flux estimates using $\nabla f$ in both experiments of assimilate OCO2+insitu or insitu only.

We hope that your major concerns are clarified and addressed. Note that we have made large revision on this manuscript that some parts of this paper are deleted or replaced (e.g., results section). Some sentences you commented on may be deleted. We are sorry about that, but these modifications are mainly based on your major concerns.

**1. Incomplete information**

Many important pieces of information are missing from the manuscript, which makes it difficult to follow and fully understand the experiments and results. Sometimes the reader is referred to previous papers (e.g., Liu et al. (2019, 2022) and Kang et al. (2012)), and at other times it is not clear where to find the missing information. I am not saying that the authors need to explain everything in this manuscript— for example, I think it is perfectly fine to refer to other papers for details about the LETKF algorithm— but there are many cases where the lack of information hinders the reader's comprehension of the present study. Often this information can be provided in one or two short sentences, so text length should not be a problem. Here are just some examples of information that I miss in the current manuscript:

1.  Ensemble size, and if the results are sensitive to ensemble size.

    Reply: Thanks for the comment. We have added the details. The ensemble size used in COLA is 20 (Line 96). It has been tested in Liu et al. (2019). And this setup is also tested in real data assimilation experiments (e.g., the OCO2MIP-v10).

2. What is the background (both fluxes and concentrations) at t=0 is. How the ensemble members were perturbed at t=0 (i.e., what is the assumed initial background error covariances?).

Reply: Thanks for the comment. We have added the details. The ensemble initial conditions for the assimilation run are randomly selected from the control run from 15 August to 15 September 2014 (Line 190). This is similar to the additive inflation process used in COLA.

3. The time resolution of the assimilated observations. The text says "For satellite data, we used a 10-second averaged …", but I assume that the system did not actually assimilate 10-second satellite observations? Were the observations averaged to daily values before being assimilated? For in situ observations, were observations from all hours used, or only e.g. measurements that were identified as being representative of large well-mixed air masses?

Reply: Thanks for the comment. We have added the details of pseudo observations to the appendix section (Text A2, Line 358). We discussed and decided not to put the details in the main text because it is not the key points of this paper and may distract readers.

The time resolution of the postprocessed real OCO-2 observation is 10-second. Since the timestep of GEOS-Chem is configured to 30 minutes, we can not sample from the nature run at 10-second timestep. Thus, the pseudo OCO-2 observation is sampled at 1-hourly timestep (Line 371) but using the real location of OCO-2 at 10-second timestep.

The insitu observations are averaged to 1-hourly at each station, and only the data from 14:00 to 16:00 local solar time are used (Line 360). The insitu network are selected from the GV obspack that are not affected by local sources (Fig. A1a).

4. The manuscript should be explicit about OCO-2 observations being column-integrated CO2 (XCO2) and explain how the synthetic XCO2 observations were calculated from the truth simulation.

Reply: Thanks for the suggestion. We have added the details of the generation of $XCO_2$ observations (Line 374, Equation 13).

5. The text refers to Liu et al. (2022) for information about the observation errors, but it would be nice if a short description could be provided here as well.

Reply: Thanks for the suggestion. The measurement error is obtained from the ObsPack. And the representative error is created based on half the variance of the deseasonalized and detrended $CO_2$ time series at a given station (Line 363, Chevallier, 2010).

Reference: Chevallier et al., 2010, $CO_2$ surface fluxes at grid point scale estimated from a global 21 year reanalysis of atmospheric measurements.

6. The inflation method is not sufficiently explained. I assume the mean of the additive perturbations were subtracted to not influence the ensemble mean (similar to what was done in Liu et al. (2022))? Were the magnitudes of the perturbations also scaled to a predefined value, and if so, what is the value?

Reply: Thanks for the suggestion. We have added the details of inflation method to the appendix section (Text A1, Line 313). Also, we discussed and decided not to put the details in the main text because it is not the key points of this paper and may distract readers. Yes, the mean of the additive perturbations was subtracted.

Similar to Liu et al. (2022), we maintained the additive inflation strategy but from an adaptive perspective. The ensemble uncertainty reduction differs in different regions with different spatial coverage of $CO_2$ observations. Typically, the reduction is expected to be small over poorly observed tropics. Thus, adding randomly selected fields to the ensembles directly may override the ensemble

characteristic and overinflate the ensemble spread in those areas. Thus, we predefined a time varying ensemble spread scale. This scale is calculated based on the temporal variability of the a priori $F_{TA}$ (SiB4). The added anomaly fields are scaled before added to ensure that the inflated ensemble spread does not greater than the predefined scale. More details can be found in the Text A1.

7. The text mentions a short (1 day) and a long (7 days) assimilation window. How were the parameters adjusted within the long assimilation window? Liu et al. (2022) states that the parameters were adjusted on a daily basis within the long assimilation window— if it is the same in this study, it should be explicitly stated. Or are the parameters assumed to be constant within the assimilation window? What are the temporal error correlations for the parameters within the same assimilation window, and do they allow the parameters to be adjusted differently on a daily basis?

Reply: Thanks for the suggestion. We have added more details on the window setup in the Method section. And we explictly guide readers to read more about the setup in Liu et al. (2019). The parameters are assumed to be constant within the long observation window. Thus, the parameters of different days within the observation window are identical.

The text decribe the windows are (Line 94): *"COLA implements a new version of LETKF with an advanced feature of a short assimilation window (1 day) and a long observation window (7 days) to enhance the SCF estimation (Liu et al., 2019). In this approach, the persistent flux parameter and dynamic $CO_2$ state are updated on a daily basis using observations within the long observation window (7 days). To achieve that, for each analysis cycle, GEOS-Chem forecasts for 7 days to generate forecast observations, which are then assimilated with the corresponding observations using LETKF. This optimization process updates the model state (CO2) and parameters (SCFs) at the end of the assimilation window, serving as the forecast initial conditions and timing for the subsequent analysis cycle. A comprehensive description of this unique LETKF feature can be found in the work of (Liu et al., 2019)."*

8. What is the temporal resolution of the flux products? From my understanding VEGAS can simulate daily fluxes, but what about CASA, are those monthly fluxes? If so, how were they downscaled to daily values?

Reply: Thanks for the comment. We are sorry that we missed the information. We have added the details. VEGAS can simulate daily fluxes. Before you mentioned the temporal resolution of CASA, I did not know that CASA can only simulate monthly fluxes, and it is downscaled to daily values based on meteorology data. In the last version we downloaded the preprocessed daily CASA data from GEOS-Chem official website. Considering that CASA is very different from VEGAS and is not a DGVM but a satellite driven model, we replaced the CASA with SiB4 as the a priori $F_{TA}$ used in this study. And SiB4 can simulate the daily fluxes.

The revised text contains the temporal information of a priori $F_{TA}$ (Line 185):" *Another terrestrial model of SiB4 (Haynes et al., 2019) is used to provide an independent daily bottom-up $F_{TA}$ estimation for the a priori regularization in the assimilation experiments*".

9. Were the fluxes/spatial gradients of fluxes assimilated before or after the CO2 observations? Does the order matter?

Reply: Thanks for the comment. The fluxes/spatial gradients of fluxes are assimilated after the $CO_2$ observations. We have revised the text and Figure 1 to show the order. Since both are treated as observations, we separated into two steps in order to assess the impact of the a priori on ensemble

uncertainty reduction (Fig. R1). We have tested to assimilate the a priori simultaneously or after, the order does not influence the results.

[Figure]

**Figure R1: a-c) The ensemble uncertainty reduction in EXP-GOI from the satellite and in-situ CO₂ observations, the a priori of the gradient, and the sum of the former two. d-f) The ensemble uncertainty reduction in EXP-FOI from the satellite and in-situ CO₂ observations, the a priori of the original flux, and the sum of the former two. g-i) The ensemble uncertainty reduction in EXP-GOI from the in-situ CO₂ observations, the a priori of the gradient, and the sum of the former two.**

This is not an exhaustive list, and I do not think it should be the reviewers' job to list everything that should be covered. The authors should think carefully about what information is relevant to include in the manuscript.

Reply: Thanks for your careful reading and we are sorry for the inconvenience. We have added other necessary information to support the key points.

*References*

Kang, J.-S., E. Kalnay, T. Miyoshi, J. Liu, and I. Fung (2012): Estimation of surface carbon fluxes with an advanced data assimilation methodology. Geophys. Res. Atmospheres, 117, D24101, doi:10.1029/2012JD018259.

Liu, Y., E. Kalnay, N. Zeng, G. Asrar, Z. Chen, and B. Jia (2019): Estimating surface carbon fluxes based on a local ensemble transform Kalman filter with a short assimilation window and a long observation window: an observing system simulation experiment test in GEOS-Chem 10.1. Geosci. Model Dev., 12, 2899–2914, doi:10.5194/gmd-12-2899-2019.

Liu, Z., N. Zeng, Y. Liu, E. Kalnay, G. Asrar, B. Wu, Q. Cai, D. Liu, and P. Han (2022): Improving the joint estimation of CO2 and surface carbon fluxes using a constrained ensemble Kalman filter in COLA (v1.0). Geosci Model Dev, 18, doi:10.5194/gmd-15-5511-2022.

**2. Confusing use of terms**

There are several terms that are used in a way that is confusing or add little to no value. Again, here are just some examples:

1. "Unique" is used throughout the manuscript, for example, "as a unique observation" (Line 21), "a unique feature of a short assimilation window" (Line 80), and "Another unique feature" (Line 82). I do not see the point of stating that these features are unique. For assimilating a priori information as observations, this has been done before (a recent example is Kaminski et al., 2022). For the first case, maybe the authors meant "as a special observation" or something similar? Similar to the first point, "comprehensive" is often used (Lines 22, 59, 111, 292), but it is not clear in what sense the described objects are comprehensive.

Reply: Thanks for the comment. We have revised the corresponding words through the manuscripts. The "unique observation" is revised to "special observation" through the manuscript (e.g., Title). The "a unique feature of a short assimilation window" is revised to "an advanced feature of a short assimilation window" (Line 89). "Another unique feature" is revised to "Another notable feature".

Thanks for pointing out that there is other similar approach. We have presented the AAPO method at the OCO2MIP telecon and Dr. Andy Jacobson also mentioned that the TransCom studies have been added the a priori flux into the observation vector. Thus, it is indeed not a unique method. And using "special" is more appropriate.

We use "comprehensive" in the manuscript to emphasis that the AAPO method also using the similar data assimilation approach as assimilating the $CO_2$ observations. Thus, both a priori and $CO_2$ observations are assimilated. We think using "comprehensive" may mislead the readers or add no value, and we deleted the words correspondingly.

2. I found it confusing that fluxes from the CASA model are described as a priori fluxes even though they were not used as a prior in the inversions. Maybe refer to them as "bottom-up fluxes" instead?

Reply: Thanks for the comment. The CASA (now SiB4) modeled fluxes is used as a priori in some experiments (EXP-FOI and EXP-FI). And the two experiments are compared with the other two experiments (EXP-GOI and EXP-GI) that used the spatial gradient of SiB4 in order to show the advantages of spatial gradient.

3. Many acronyms are confusing. For example, in Summary and Discussion, AAPO is redefined on Line 290, but it is not clear in this context how the acronym was formed.

Many equations introduce new acronyms, for example TSG and PSG in Equation (5), and ASG in Equation (8). How about using a different variable (e.g., $\nabla f$) or a subscript (e.g., "SG") for the spatial gradient of the fluxes, and then superscripts A, T, and P for analysis, truth, and prior, respectively? This would reduce the number of acronyms the reader has to keep track of.

Equation (6) introduces the "S" superscript, which I assume stands for "SCF", but it is not defined or used anywhere else in the manuscript, which is confusing. I would suggest dropping the superscript, or have "SG" and "f" as subscripts in Equations (5) and (6) instead. Generally, try to be consistent throughout the manuscript.

Similarly, what do "NP", "P", "ASG", and "AP" in the experiment names stand for? (I have my guesses, but it should not be the reader's job to figure this out.)

Reply: Many thanks for the constructive suggestion. The acronyms are misleading and not informative. We have deleted some unnecessary equations and carefully revised the acronyms in the equations. We have checked the acronyms that are redefined or not clearly defined.

The spatial gradient of flux is represented by ∇**f**, while the flux itself is represented by **f** (Line 205, Equation 3, 4). The truth, analysis, and a priori are represented by T, a, and Ap, respectively. The meaning of experiment names is changed to represent the a priori and $CO_2$ observations and described in section 3.1. For example, EXP-GOI stands for assimilates the Gradient and the synthetic OCO-2 and in-situ observation (Line 191, Table 1).

4. Line 138 says that "COLA is a flow-dependent ensemble-based DA". The term "flow-dependent" makes sense in the context of e.g. atmospheric data assimilation, where the errors depend (and often follow) the atmospheric flow, but it is not clear what exactly it means here. Given that there is no dynamic model for the parameters, how is the system flow-dependent?

Reply: Thanks for the comment. We use "flow-dependent" to emphasis that COLA fully uses the persistence from the last assimilation time as the first guess of the current assimilation time. This may mislead readers and is not appropriate. And I agree that flow-dependent make sense with dynamic model. Thus, we deleted the corresponding expression.

5. The text uses "significant/significantly" in many places. I would recommend reserving these words for describing statistical significance when comparing results. If the authors performed statistical significance testing, they should state what tests were used and the significance level.

Reply: Thanks for the suggestions. We have made large revision on the results. Following your suggestions, we avoid using "significant/significantly" when comparing two experiments but using ratio/bias to show the difference and improvement. For example, in Line 280, "*In the North American Temperate and Europe, where the observations are dense, ..., and the RMSEs in EXP-FOI are larger than those in EXP-OI by 59.2% to 91.5%*".

6. Lines 289–290 state that "the spatial gradient of a bottom-up model estimation is dynamically assimilated". What is "dynamically" referring to here?

Rely: Thanks for the comment. In this context, we use "dynamically" to emphasis that the spatial gradient calculation can partly compensate the deficiency of lacking a dynamic model. This is not an accurate expression. Thus, we deleted "dynamic/dynamically" in the title and main text corresponding.

*References*

Kaminski, T., M. Scholze, P. Rayner, M. Voßbeck, M. Buchwitz, M. Reuter, W. Knorr, H. Chen, A. Agusti-Panareda, A. Löscher, and Y. Meijer (2022): Assimilation of atmospheric CO2 observations from space can support national CO2 emission inventories. Environmental Research Letters, 17, 014015, doi:10.1088/1748-9326/ac3cea.

**3. Language and figures**

The language could be improved in several instances, for example:

1. Articles are sometimes missing or incorrectly added, or the noun should be plural instead of singular or vice versa. Here are just a few examples: "and a priori flux from …" -> "and a priori fluxes from" (Line 15), "weigh a priori flux to the background" -> "weigh a priori fluxes to the background" (Line 16), "because of the deficiencies in understanding" -> "because of deficiencies in understanding" (Line 19), "LETKF estimates SCF as …" -> "LETKF estimates SCFs as …" (Line 76).

Rely: Thanks for finding the mistakes. We have revised the corresponding words you mentioned and other mistakes throughout the manuscript. And it is polished by AJE.

2. The text often uses subjective words such as "good" and "better". I would suggest using more objective and descriptive words. For example, instead of saying "produces much better

estimation …", I would say something like "produces more accurate flux estimates in terms of x and y".

Reply: Thanks for the suggestions. In the revised results section, the comparison between experiments are not described by descriptive words. For example, in Line 261, "*However, the global RMSE in EXP-GOI is 26.4% smaller than that in EXP-OI, indicating that the gradient is better than the original flux in estimating the global seasonal cycle*".

3. When describing what was done, I suggest using past tense ("GEOS-Chem was run" (Line 160), "Four experiments were performed" (Line 205), etc.) rather than present tense.

Reply: Thanks for the suggestions. We have checked and revised the tenses.

The figures are not the clearest. Here are a few suggestions for improvements:

1. Figure 2: The blue-red colorbar has similar lightness at the upper/lower ranges of the colorbar, especially for the red part, which makes it hard to see the difference between e.g. 2.0 and 1.6. Maybe use a discrete colorbar, and change to a colormap with a larger range of lightness.

Reply: Thanks for the suggestions. We have revised the colormaps in map figures with more and clearer levels. In Figure 2, the colormap is changed from blue/white/red to blue/skyblue/white/orange/red.

2. Figure 3: It is sometimes hard to discern the "truth" line in the top panel. Maybe make the black line thicker. Also Figure 3: It would be easier to read the plot if the RMSE/BIAS values were listed in the same order as the legend showing the experiments.

Reply: Thanks for the suggestions. In Figure 3 (now Figure 4), we have made the "truth" line thicker. The RMSEs are presented following the order of legend.

**4. Concerns about the experimental setup**

I have several concerns about the experimental setup, which could affect the interpretation of the results. My main concern is that the assumptions are too idealistic to be applicable in real data cases. In particular, the OCO-2 observations in these perfect model OSSEs will directly reflect the surface CO2 fluxes and therefore provide a strong constraint on the spatiotemporal flux variations. In the real world, XCO2 observations are generally sensitive to other error sources such as systematic errors, representativeness errors (the swath width of OCO-2 is much smaller than the model grid points here and the satellite measurements are also affected by e.g. clouds), errors in the atmospheric CO2 background, and atmospheric transport errors. As a first step, how were representativeness errors included in the synthetic observations? Did the authors assume that there was an XCO2 observation in a 4°×5° model grid cell if there were more than x% XCO2 measurements in the grid cell? If so, what was the x threshold?

Reply: Thanks for the comment. We understand your concern about the observations. Since this study is aimed at validating the AAPO method in the context of OSSE, the effect of AAPO may be mixed with other factors if we consider other sources of uncertainty/error. And the systematic error of satellite retrievals is not fully understood currently (Taylor et al., 2023), it is hard for us to add such error to the pseudo observations. However, the generation of pseudo OCO-2 observations do add a random error to it (Text A2). The pseudo OCO-2 retrieval is created based on the following equation,

$$XCO_2^{pseudo} = \mathbf{p} \cdot ((\mathbf{I} - \mathbf{ak}) \cdot \mathbf{c}^{prior} + \mathbf{ak} \cdot \mathbf{c}^{nature}) + \varepsilon,$$

where $XCO_2^{pseudo}$ represents a pseudo OCO-2 retrieval; $\mathbf{p}$ is the vertical pressure weight profile corresponding to the averaging kernel profile $\mathbf{ak}$; $\mathbf{c}^{nature}$ is the a priori $CO_2$ vertical profile prescribed by the ACOS retrieval algorithm; $\mathbf{c}^{nature}$ is the simulated $CO_2$ vertical profile sampled from the nature run; $\mathbf{I}$ is an identical vector; and $\varepsilon$ is the added zero-mean random uncertainty based on the observation error. Baker et al. (2022) described how the error is estimated during the 10-second averaging process.

Since the OCO-2 data provided by Baker et al. (2022) is a 10-second averaged product, and it has been exstensively used in the OCO2MIP inversions. The distance between two adjacent observation is ~67.5 km (~0.6 degree). Thus, there are approximately 6 observations within a model grid cell, and there are around 80 observations in some areas annually (Fig. A1b). Following the "assimilate_flag" described in postprocessed OCO-2 data, only good quality observations in the real data are used to generate the pseudo observation in the OSSE.

We understand your concerns, since the spatial resolution of OCO-2 is very small compared with model resolution, it is not practical to fully reproduce the characteristics of the real observation. But the added random errors are independent from each other. And, similar approach of generating pseudo satellite observation has been applied in published researches to exam their inversion systems (e.g., Liu et al., 2014; )

References:

*TE Taylor et al., 2023, AMTD, Evaluating the consistency between OCO-2 and OCO-3 XCO estimates derived from the NASA ACOS version 10 retrieval algorithm.*

*DF Baker et al., 2022, GMD, A new exponentially decaying error correlation model for assimilating OCO-2 column-average $CO_2$ data using a length scale computed from airborne lidar measurements.*

To make it easier to understand the results and compare them with other established inversion systems, I suggest the authors perform another set of OSSEs where only in situ observations are assimilated.

Reply: Thanks for the suggestion. We have made two extra experiments (EXP-FI and EXP-GI) that only assimilate the insitu observations. Comparing between the two experiments, we find that the performance of EXP-GI that using the spatial gradient is also better than that in EXP-FI.

I am also wondering if some of the results could be due to the specific algorithms in COLA rather than indicative of the performance of general inversion systems. For example, the uncertainty of the assimilated bottom-up fluxes/flux gradients were assumed to be proportional to the analysis uncertainty. Thus, if I understand this correctly, the data assimilation system will place more weight on the assimilated bottom-up fluxes/flux gradients compared with the CO2/XCO2 observations the more confident it is in its analysis. Is this a reasonable assumption? It seems to me that this would bias the EXP-P experiment toward the bottom-up fluxes, which seems to be what happens if you look at Figure 4, given that the analysis uncertainty will likely be reduced a lot when assimilating direct observations of the fluxes.

Reply: Thanks for the comment. Since there are some configurations/method are only applied in COLA, we acknowledge that the results may be affected by them. For example, the window length of COLA is relatively shorter than some long window-based inversion systems. We indicated that COLA may not be bettern than those system in in-situ only inversion (Line 247).

However, the uncertainty of a priori is assumed to be proportional to but 5 times larger than the analysis uncertainty, which means that the weight of a priori is assumed to be small. And the a priori only serves to regularize the ill-posed problem and does not overwrite the information given by the $CO_2$ observations. The weight of a priori and $CO_2$ observation can be expressed by the ensemble uncertainty reduction (EUR), which is defined as follows,

$$\mathbf{EUR}_{i,t} = \frac{\sigma^b_{i,t} - \sigma^a_{i,t}}{\sigma^b_{i,t}},$$

where $\sigma^b_{i,t}$ and $\sigma^a_{i,t}$ is the first guess and final analysis ensemble uncertainty, respectively, at a given grid point i and a given time t. Since there are two types of observation, the EUR can be separated to two parts of $EUR^{CO2}$ and $EUR^{Ap}$ as,

$$EUR_{i,t}^{CO2} = \frac{\sigma_{i,t}^b - \sigma_{i,t}^{a*}}{\sigma_{i,t}^b},$$

$$EUR_{i,t}^{Ap} = \frac{\sigma_{i,t}^{a*} - \sigma_{i,t}^a}{\sigma_{i,t}^b},$$

where the superscript $a*$ denote the analysis after assimilating $CO_2$ observation; the superscript Ap denotes the a priori.

As show in Figure R1, for EXP-GOI, the $EUR^{CO2}$ in the northern middle and high latitudes of North America and Europe, where the $CO_2$ observation network is dense, can exceed 30%. Generally, the $EUR^{CO2}$ decreases from north to south. In South America and Africa, the $EUR^{CO2}$ is around 10%. Since the uncertainty of a priori is set to be proportional to the ensemble uncertainty, ***$EUR^{Ap}$ is approximately 5% and almost identical at different grids. For EXP-FOI, the $EUR^{Ap}$ is configured to be identical to EXP-GOI; thus, the weights of a priori in EXP-GOI and EXP-FOI are expected to be identical.*** However, in EXP-GOI, the $EUR^{Ap}$ in the Qinghai-Tibet Plateau, Andes, Sahel, and Arabian Plateau is relatively larger than in the other areas, suggesting that, even though the error of the gradient at different grid points is identical, the $EUR^{Ap}$ is also affected by the geography. In EXP-GI, which assimilated only the in-situ data, the $EUR^{CO2}$ is approximately half that in EXP-GOI, resulting in similar $EUR^{CO2}$ and $EUR^{Ap}$ in South America and Africa.

For the additive inflation method, given that the ensemble flux deviations and the additive flux perturbations can both be positive and negative, is it possible that the ensemble spread will actually decrease when adding the perturbations?

Reply: Thanks for the comments on the additive inflation method. We have added a section to describe the inflation method (Text A1). Yes, since the flux perturbations are randomly selected from the a priori fluxes, it is possible that the ensemble spread will decrease when adding the perturbations.

Additionally, it is also possible that the ensemble spread will be overinflated. Because the $EUR^{CO2}$ differs in different regions with different spatial coverage of $CO_2$ observations. Typically, the $EUR^{CO2}$ is expected to be small over poorly observed tropics (Fig. R1, first column). Thus, adding randomly selected fields to the ensembles directly may override the ensemble characteristic and overinflate the ensemble spread in those areas. Thus, to partly overcome the decreased or overinflated problem, a time-varying ensemble uncertainty scale based on the temporal variability of the bottom-up estimation is defined before adding those selected fields. And the perturbations are rescaled based on the scale.

Another point about the inflation method: the daily fluxes vary mostly due to varying weather and advancing seasonal cycles (this is especially true if the fluxes were downscaled from e.g. monthly values). Thus, there are likely strong temporal correlations in the daily flux variations. I wonder if way of choosing additive inflation anomalies is optimal considering for example the subspace spanned by the ensemble members after adding the inflation perturbations. A quick check could be to perform a principal component analysis on the additive ensemble perturbations from the inflation method and see how the explained variance declines.

Reply: Thanks for the comments. We understand your concerns on the additive method. Since the paper is focused on the AAPO method, adding the details may distract readers from the main point and we put the details on the appendix section (Text A1).

We agree that there are strong temporal correlations in the daily flux variation, and the daily variations are what COLA looking for. Since The first guess of the SCFs is derived based on the persistence of the analysis from the previous assimilation cycle. The error of the first guess is the flux difference

of the current and the previous assimilation time. For example, the seasonal cycle is coherently evolving in the Northern Hemisphere and the added perturbation based on the additive method could partly express the spatiotemporal correlation. Since the inflation method approximates the covariance, it does not fully reproduce the real covariance structure.

Related to the previous point, would this scheme not put less emphasis on fluxes in regions with smaller seasonal cycles and/or less variations in weather (in particular temperature and solar radiation)? Is this desirable? It would be informative if there was a figure (for example in an appendix) showing some examples of the additive inflation flux anomalies, or the leading modes and their explained variance from the principal component analysis.

Reply: Thanks for the comments. Based on the additive inflation scheme, the ensemble spread in the tropics is smaller than that in the northern extratropics. However, this is not meaning that the scheme put less emphasis on the fluxes in the tropics.

If give a large ensemble spread in the tropics, the variation of the analysis fluxes could be larger than the temporal variation of the a priori fluxes and may lead to large temporal dipole noises. In some condition, the temporal variation of the a priori flux may be smaller than the true flux. To overcome this situation, we added an annual-mean ensemble uncertainty scale to the inflation process to ensure that the ensemble spread is not underestimated (Text A1, Line 308).

**5. Comparison with traditional methods**

The manuscript claims to compare the results from the new algorithm with a "traditional" method. Here the "traditional" method is basically Equation (2), whereby fluxes are forecast forward in time based on the analysis from the previous two time steps and the bottom-up fluxes at the forecast time. However, I would not call this the traditional method as (i) to my knowledge, this method is only used by CarbonTracker/CarbonTracker Europe and maybe one or two other systems; and (ii) the implementation here is not the same as the one in CarbonTracker or in the cited reference. For one thing, the dynamic model in CarbonTracker is applied on flux parameters, which are scaling factors on bottom-up fluxes, and not on the fluxes themselves (as in this manuscript). The idea is that the bottom-up fluxes capture the high-frequency variations in the fluxes, for example due to shifting weather, while the flux parameters correct for long-term systematic errors. It may therefore be reasonable to smooth the flux parameters in time and relax them back to the prior values. In this manuscript, however, Equation (2) would partly relax the fluxes back to the previous days' values, which would likely degrade the forecast fluxes in most cases (considering the advancement of the seasonal cycle and movement of weather systems) compared with the persistence model.

Reply: Thanks for pointing out the mistakes on describing the methods applied in CarbonTracker and CarbonTracker-EU. We have revised the corresponding sentences (Line 114): "*Consequently, in CO$_2$ inversion systems with longer assimilation windows, the initial estimate of carbon fluxes heavily relies on a priori regularization techniques. For example, Feng et al. (2009) utilized the a priori information as the initial guess for their EnKF calculations, while Peters et al. (2007) employed the a priori to derive carbon flux variation modes and corresponding scaling factors, which are optimized through EnKF*".

In the scheme of COLA (Line 109), the first guess of the SCFs is derived based on the persistence of the analysis from the previous assimilation cycle, which is considered a reasonable approximation for the current SCF estimate. This assumption of persistence aligns well with COLA's short assimilation window of 1 day. However, for systems with longer assimilation windows, such as those spanning several weeks for SCF estimation, the temporal dynamics of the terrestrial ecosystem,

including diurnal cycles and the transition from winter respiration to rapid spring growth, can introduce temporal lag and inaccuracies. Consequently, in $CO_2$ inversion systems with longer assimilation windows, the initial estimate of carbon fluxes relies more on a priori fluxes.

In conclusion, the persistence scheme applied in COLA relies on the short assimilation window of 1 day. As the reviewer pointed out that, if the window is long, the persistence is definitely not a good assumption and may lead to temporal lag.

This together with the concerns raised in the previous points makes it unsuitable in my opinion to refer to EXP-P as using "traditional" methods. I do agree that there should be a comparison with established CO2 inversion methods to assess the added value of the new algorithms introduced in this paper. Thus, I recommend the authors to set up a more "standard" inversion experiment and compare the results from this inversion with their results.

Reply: Thanks for the comments. We agree that EXP-P is not really a "Traditional" method. Since COLA is different from other traditional systems in the assimilation schemes, it is hard for us to make an experiment using real 'traditional' methods.

Thus, to make the key point clear, we decide to delete the experiment that directly add a priori fluxes to the first guess because it is not comparable to the experiment using the AAPO scheme in terms of the weight of a priori. And the experiments using AAPO methods modified each ensemble member and reduce uncertainty (Fig. R1), while EXP-P only modified the ensemble mean. Finally, the key point of this paper is focusing on the comparison between the flux and its spatial gradient.

**Specific comments**

Lines 18–19: "systematically biased at different spatiotemporal scales because of the deficiencies in understanding of some underlying processes". It could also be due to uncertain parameter values and/or initial conditions in variables related to the carbon cycle.

Reply: Thanks for the comments. We have revised to, "*However, the "bottom-up" flux estimations, especially the simulated terrestrial-atmosphere $CO_2$ exchange, are usually systematically biased at different spatiotemporal scales due to the limitation in understanding and parameterizing some underlying processes.*".

Lines 26–28: "We suggest that the AAPO algorithm can be applied to other greenhouse gas (e.g., CH4, NO2) and pollutant data assimilation studies." This was not shown in the study, so I suggest removing it from the abstract.

Reply: Thanks for the comments. We have deleted this sentence in the abstract.

Lines 35–36: "of the Bayesian synthesis (…) and data assimilation (DA) techniques". I wonder why the authors separate between Bayesian synthesis and data assimilation techniques here. Most data assimilation methods in widespread use are based on Bayesian inference (Rayner et al., 2019).

*References*

Rayner, P. J., A. M. Michalak, and F. Chevallier (2019): Fundamentals of data assimilation applied to biogeochemistry. Atmos. Chem. Phys., 19, 13911–13932, doi:10.5194/acp-19-13911-2019.

Reply: Thanks for the comments. We agree that this may lead misunderstandings. The sentence is revised to, "*The SCFs can be inferred from atmospheric $CO_2$ measurements using "top-down" (hereafter quotation marks will be omitted) techniques of the Bayesian synthesis (e.g., Rodenbeck et al., 2003; Zammit-Mangion et al., 2022; Cho et al., 2022) and, ensemble Kalman filter (e.g., Peters et al., 2007; Feng et al., 2009; Jiang et al., 2022; Liu et al., 2022), and 4-dimentional variational (4D-var) techniques (Chevallier et al., 2010; Liu et al., 2014).*"

Lines 37–28: "However, the top-down estimation could be ill-posed because of … and systematic errors of the transport model and satellite retrieval". I would argue that the most common reason for ill-posed problems in inverse modeling is that there is not a unique solution, which is mainly caused by a lack of observational constraint and insufficient regularization. I would not consider systematic errors to be the primarily cause of ill-posed problems. From a quick scan I could not find the term "ill-posed" mentioned in any of the cited references.

Reply: Thanks for the comments and suggestion. We agree that the systematic errors are not the reason of being "ill-posed" but other influence factors. The sentences are revised to be clearer in Line 41, "*However, the top-down estimation could be ill-posed because of the sparseness feature of atmospheric $CO_2$ observations. And systematic errors in the transport model and satellite retrieval can contaminate the illustration of inferred SCFs (Basu et al., 2018; O'Dell et al., 2018; Yu et al., 2018; Schuh et al., 2019)*".

Lines 66–67: "The COLA DA system consists of …, and the assimilated observations". I would not include the assimilated observations in the COLA DA system (by this logic, you should also include the meteorological driving data, the bottom-up fluxes, etc.).

Reply: Thanks for the comments and suggestion. We have deleted the "assimilated observations" in this context (Line 74). The meteorological and flux data are forcings that adding them to this sentence maybe too complicate. Thus, we prefer to describe them independently (Line 77).

Lines 76 and 91: "LETKF estimates SCF as evolving parameters" and "COLA treats SCFs as stationary parameters". I get what the authors mean, but for readers who are not familiar with the method, this could sound contradictory.

Reply: Thanks for the comments and suggestion. SCFs are evolving parameters between assimilation windows but stay stationary within the assimilation/observation window. The sentences are revised in Line 84, "*In the LETKF algorithm, the SCFs are treated as evolving parameters between assimilation windows*", and in Line 107, "*the SCFs are treated as stationary parameters within the observation window, meaning that they remain constant during model forecasting and are updated exclusively using the LETKF algorithm*".

Lines 76–77: "by augmenting it with the state vector CO2". I would rather say that "the state vector, which contains atmospheric CO2 concentrations, was augmented to also include the SCF parameters" or something to that effect.

Reply: Thanks for the comment and suggestion. The sentence is revised in Line 85, "*The state vector, which contains atmospheric $CO_2$ concentrations, is augmented to also include the SCFs parameters*".

Line 77: "the LETKF prefers a short assimilation window". I would argue that it depends on the application, and in some applications an ensemble Kalman Smoother with a longer assimilation window is more desirable.

Reply: Thanks for the comment and suggestion. We agree that it depends on the application. But, in this context, we meant for state data assimilation, for example in weather data assimilation and $CO_2$ state data assimilation. And after this sentence, we further argue that the flux parameter benefits from a long period of training (Line 88) that an ensemble Kalman Smoother with a longer assimilation window is more desirable.

Line 94: "There are two widely used a priori regularization approaches for EnKF-based carbon inversion systems." As previously mentioned, I believe this is misrepresenting the regularization approaches commonly used in inversions.

Reply: Thanks for the comment. Instead of referring them as traditional method, we just described these methods but not saying them as "traditional" and have revised the sentence to be more accurate, Line 111, "*However, for systems with longer assimilation windows, such as those spanning several weeks for SCF estimation, the temporal dynamics of the terrestrial ecosystem, including diurnal cycles and the transition from winter respiration to rapid spring growth, can introduce temporal lag and inaccuracies. Consequently, in $CO_2$ inversion systems with longer assimilation windows, the initial estimate of carbon fluxes heavily relies on a priori regularization techniques. For example, Feng et al. (2009) utilized the a priori information as the initial guess for their EnKF calculations, while Peters et al. (2007) employed the a priori to derive carbon flux variation modes and corresponding scaling factors, which are optimized through EnKF*".

Lines 98–99: "This approach omits useful information on the temporal dependency of SCFs". I do not understand the authors' point here—how does this omit information on the temporal dependence, given that the temporal variations are captured by the bottom-up fluxes?

Reply: Thanks for the comment. We are sorry that we misunderstand the method. Yes, the bottom-up fluxes provide the temporal variations. We have revised the description in Line 111.

Line 106: "we propose a new a priori regularization method to better follow the DA principle". The DA principle, as I see it, is to optimally combine a priori information with observations. I do not see how this new regularization method better follows the data assimilation principle, considering that it does not use an informative prior for the fluxes.

Reply: Thanks for the comment. This description is not accurate, and we revised it in Line 128, "*Based on the DA principle, we propose a novel a priori regularization method for COLA*".

Lines 127–128: "we define the SCF spatial gradient at a given grid as the SCF difference with its surrounding grids divided by the distance between them." Is this considering only grid points orthogonally adjacent, or also diagonally adjacent (I assume the former)? The gradient of a 2D field has two components, in this example x and y, but this is not mentioned anywhere. I suspect that the authors accounted for this given that Equations (5) and (8) have an overhead arrow for the gradient variables, but the meaning of the arrow is not mentioned anywhere. Additionally, the vertical bars are not defined—I assume they are used to denote the Euclidean norm?

Reply: Thanks for pointing out the missing information. Yes, we only considering the orthogonally adjacent grid points. We revised the sentence in Line 153, "*Numerically, we define the $\nabla f$ vector at a given grid as the SCFs difference with its orthogonally adjacent grids divided by the distance between them. Thus, there are two components in the $\nabla f$ along the longitude and latitude direction*".

In Equation 3 and 4, the vertical bar is the Euclidean norm. We have added the details in Line 207.

Lines 168–170: "A nature run is driven by the $F_{OA}$ from Rödenbeck et al. (2014), $F_{FE}$ from the Open-source Data Inventory of Anthropogenic $CO_2$ emissions (ODIAC) (Oda et al., 2018), and the $F_{IR}$ ..." Do these other flux components (ocean, fossil fuel, and fire) matter if they are identical in the truth and inversion experiments (considering that the atmospheric transport is perfect)?

Reply: Thanks for the comment. Since the transport is perfect, the other components do not matter.

Equations (7)–(9): Do t1 and t2 represent the actual time (as the text suggests), or are those time steps? If it is the latter, why was t2-t1 used rather than t2-t1+1? (The latter would correspond to division by the number of samples.)

Figure 3: The lower panel of Figure 3 shows that the RMSE and bias of EXP-NP are equal (1.36 GtC/yr). Normally in such a situation I would think that almost all of the RMSE can be explained by the bias, but this is not the case when looking at the lines in Figure 3. Do the RMSE and bias happen to be the same

here because the denominator uses N-1 rather than N (where N is the number of samples) in the definitions of RMSE and MB (see previous point)?

Reply: Thanks for pointing out the mistake. t1 and t2 represent the timestep. We have revised the Equation 2 and the description for the equation.

**3. Technical corrections**

Line 76: Remove parentheses around citation.

Reply: Thanks for pointing out the mistake. We have deleted the parentheses.

Line 77: "Similar to the other EnKF" -> "Similar to other EnKF variations"

Reply: Thanks for the suggestion. We have revised the corresponding sentence in Line 86.

Line 79: "long training period". Do the authors mean "a long assimilation window"? "Training" is not defined in the manuscript.

Reply: Thanks for the comment. "Training" may mislead readers. We replaced the "long training period" to "long observation window".

Line 97: "and the subscripts b, p, and t" -> "and the superscripts b and p and subscript t"

Reply: Thanks for pointing out the mistake. Based on your previous major comments, we have deleted the corresponding descriptions.

Figures 6 and 7: Change "Kg" to "kg".

Reply: Thanks for the suggestion. We have changed "Kg" to "kg" in Figure 3.

Line 232: "show an increase in RMSE of FTA concerning EXP-NP" Do the authors mean "compared with" instead of "concerning"?

Reply: Thanks for the suggestion. Yes, it means "compared with". Since we have made a large revision on the results section. The corresponding sentence was deleted. And we are sorry about this.

As mentioned there could be other issues that I have not raised here, which can be properly assessed only after the authors have included the missing information and clarified the manuscript.

Reply: Thanks for the careful readings. We have careful revised other sentences and hope that we have addressed your major concerns.

**Table 1: Assimilation experiments setup.**

| Experiment | EXP-GOI | EXP-FOI | EXP-OI | EXP-GI | EXP-FI |
|---|---|---|---|---|---|
| A priori | $\nabla f$ | $f$ | | $\nabla f$ | $f$ |
| Observation | OCO-2+In-situ | OCO-2+In-situ | OCO-2+In-situ | In-situ | In-situ |

[Figure]

**Figure 1: The assimilation cycle of the COLA system, illustrating how and where the a priori is assimilated.**

[Figure]

**Figure 2: a) The annual mean signal-to-noise ratio pattern of the a priori $F_{TA}$ spatial gradient. b) The same as a) but for the a priori $F_{TA}$. c) Annual mean signal-to-noise ratio of the spatial gradient divided by the signal-to-noise ratio of the $F_{TA}$.**

[Figure]

**Figure 3: The annual mean F$_{TA}$ in 2015 of a) the truth, b) the a priori, c) EXP-GOI, d) EXP-FOI, e) EXP-GI, and f) EXP-FI. g) Comparison of the annual total F$_{TA}$ in the northern extratropical area and the tropical and southern extratropical areas. Different scatters denote the truth, the a priori, and the different experiments. The dashed black line denotes the global carbon budget.**

[Figure]

**Figure 4: The top figures in each subplot are the a) global, b) northern extratropical, and c) tropical and southern extratropical seasonal cycles of F$_{TA}$ in the truth (black), the a priori (gray), EXP-GOI (red), EXP-FOI (orange), and EXP-OI (sky blue). The bottom figures in each subplot are the global total difference of the a priori and the three assimilation experiments compared to the truth. The annual mean RMSEs of the a priori F$_{TA}$ and the three experiments are denoted at the upper right corner of the bottom figures.**

[Figure]

**Figure 5: The regional RMSE of the a priori and the five assimilation experiments compared to the truth. The regions are defined by the OCO2MIP.**

---

## Author Comment (AC2)

**Reply to reviewer 2:**

In this manuscript, an alternative a priori flux constraint is presented in the context of a global CO2 flux inversion performed using an ensemble Kalman filter (EnKF) with a short assimilation window. Observing system simulation studies (OSSEs) are preformed to give an idea of how this alternative constraint might function when used with real data in a real inversion. The flavor of EnKF used is the local ensemble transform Kalman filter (LETKF), as implemented in the Carbon in Ocean–Land–Atmosphere (COLA) data assimilation system, a global CO2 flux inversion based on the GEOS-Chem transport model.

The alternative flux constraint is formulated in terms of the spatial gradient of the fluxes: finite differences of flux using adjacent grid boxes in the model. These spatial gradients are then added as new measurements in the measurement vector, as opposed to additional constraints in the traditional a priori state vector. Gradients used in this manner could capture the bulk of the flux constraint (its spatial and temporal patterns), while at the same time cutting the tie to the absolute value of the flux -- i.e. its overall constant offset or long-term mean. This in turn could be useful when using priors for which the variability is more robust than the long-term mean -- for example, the terrestrial biosphere models used as priors for CO2 fluxes over land in global flux inversions, which do a good job getting the seasonality of the fluxes right (e.g., using satellite measurements vegetation greenness, plus assumptions on the timing of respiration) but a less-good job of estimating the integrated flux across a full year. By getting rid of the constraint to the long-term mean of the prior, the flux estimate might be freer to move to the long-term mean given by the data and not suffer from being biased in the direction of the incorrect or inaccurate prior. This of course would be at the cost of losing any benefit that that long-term prior mean might provide. In general, a flux constraint of this nature should be able to be implemented as a measurement in the measurement vector, as is done here, assuming that the measurement uncertainty used gives the constraint the same weight as it would have had if it had been implemented more traditionally in the a priori state vector. One would have to avoid double counting by not also having the traditional flux prior in force at the same time.

In their OSSE experiments, the authors compare the effectiveness of this flux spatial gradient constraint against the usual prior flux constraint (i.e. in terms of the actual flux value itself, not the spatial gradient) implemented either in the measurement vector or, more traditionally, as part of the a priori state vector; in the latter case, a couple different forms for the first guess of the flux at the new measurement time are used: either 1) a combination of the prior flux at the given time plus the flux estimate from the EnKF at the two immediately-earlier times, or 2) just the prior flux at the new time. This is done using one land biospheric model (VEGAS) to generate the 'true' measurements, and a second model (CASA) to be used as the prior flux. The authors find that, in general, when the flux gradient prior is used, the EnKF does a better job estimating the true fluxes than when three other approaches based on the absolute fluxes themselves (i.e., not gradients) are used.

While these results look promising, there are some inconsistencies in the results that I would like explained. Also, I suggest modified OSSEs in which the ocean fluxes are allowed to be corrected along with the land fluxes, in order to give a more realistic test of the new constraint. Finally, there is a lack of detail in the description of the methods used that makes it difficult for me as a reviewer to assess the full meaning of the results. I suspect that the general reader will have similar questions. I suggest that the authors add these needed details to the manuscript, address the points that I raise below, and resubmit, at which point I will re-review it and decide on final publication.

Reply: Many thanks for your constructive comments/suggestions and recognizing the AAPO method. We acknowledge that we have missed some important information and the experiments are not optimal. We are sorry for the inconvenience. We have reworked the manuscript in terms of method, experiment setup, and results. And the manuscript was also polished by AJE. The main changes are as follows:

1.  Information: Adding more details about the short assimilation window and long observation window / experiment setup of initial condition, ensemble size etc. / observation localization / additive inflation / generation of pseudo observation.
2.  Clearer method description: a) Assimilating $CO_2$ observation before assimilating a priori (Fig. 1). b) Using $\nabla \mathbf{f}$ to represent the spatial gradient.
3.  New experiment setup: a) OCO2+insitu vs insitu only. b) Making sure the weight of a priori is identical in different experiments. c) Changing CASA to SiB4 as the a priori.
4.  Clearer message from the OSSE results: Better hemispheric flux estimates using $\nabla \mathbf{f}$ in both experiments of assimilate OCO2+insitu or insitu only.

We hope that your major concerns are clarified and addressed. Note that we have made large revision on this manuscript that some parts of this paper are deleted or replaced (e.g., results section). Some sentences you commented on may be deleted. We are sorry about that, but these modifications are mainly based on your major concerns.

Comments:

First, the authors should describe in detail [with equations] the meaning of the terms 'assimilation window' and observation window', since how these terms are used in the context of the LETKF is not generally known. The reader should not have to go back to the previous LETKF papers to find this. Does the 1-day assimilation window mean that the filter is stepped forward in time a day at a time, each day allowing the new measurements to update the fluxes across the 7-day measurement window (i.e. the current day plus six previous days)? If so, the weight given to the flux constraint (or flux prior constraint) for each of those 7 days ought to be reduced, so that the integrated effect of the seven days of measurement updates affecting the fluxes on a given day is equivalent to the weight given to a single days' flux prior in some other estimation method (e.g. a variational method or a matrix-inversion-based Bayesian synthesis method).

Reply: Many thanks for your suggestions. We have added descriptions on the assimilation/observation window setup (Line 89) and guide the readers to refer Liu et al. (2019) for more details.

In this window steup, the persistent flux parameter and dynamic $CO_2$ state are updated on a daily basis using observations within the long observation window (7 days). To achieve that, for each analysis cycle, GEOS-Chem forecasts for 7 days to generate forecast observations, which are then assimilated with the corresponding observations using LETKF. This optimization process updates the model state ($CO_2$) and parameters (SCFs) at the end of the assimilation window, serving as the forecast initial conditions and timing for the subsequent analysis cycle. A comprehensive description of this unique LETKF feature can be found in the work of (Liu et al., 2019).

We intend to add an equation to illustrate it, but we find it may add more questions to the readers. The LETKF analysis equation is,

$$\bar{\mathbf{x}}_{t_1}^a = \bar{\mathbf{x}}_{t_1}^b + \mathbf{X}_{t_1}^b \widetilde{\mathbf{P}}^a \left(\mathbf{Y}_{t_0 \to t_7}^b\right)^T \mathbf{R}_{t_0 \to t_7}^{-1} \left(\mathbf{y}_{t_0 \to t_7}^o - \bar{\mathbf{y}}_{t_0 \to t_7}^b\right)$$

$$\widetilde{\mathbf{P}}^a = \left[\left(\mathbf{Y}_{t_0 \to t_7}^b\right)^T \mathbf{R}_{t_0 \to t_7}^{-1} \left(\mathbf{Y}_{t_0 \to t_7}^b\right) + (K-1)\mathbf{I}\right]^{-1}$$

where the flux parameter $\mathbf{f}$ is augmented to the $CO_2$ state $\mathbf{c}$ that $\mathbf{x} = [\mathbf{c}, \mathbf{f}]^T$; the superscripts a and b denote the analysis and background (first guess), respectively; $\bar{\mathbf{x}}$ and $\mathbf{X}$ are the ensemble mean and ensemble perturbation, respectively; the subscript $t_1$ indicate the end of assimilation window of 1 day; $\mathbf{y}^o_{t_0 \to t_7}$ is the $CO_2$ observations within the observation window of 7 days; $\mathbf{y}^b_{t_0 \to t_7}$ is the forecasted observations corresponding to each observations; $\mathbf{Y}^b$ is the ensemble perturbation in the observation space; $\mathbf{R}$ is the observation error matrix; $\widetilde{\mathbf{P}}^a$ is the analysis error covariance; K is the ensemble size which is set to 20; and $\mathbf{I}$ is the identical matrix.

In the LETKF analysis equation, the windows are expressed by the subscripts, readers may misunderstand that there are only 7 timesteps of observations. We have discussed for several time and decide not putting it in the manuscript but describing it directly.

Second, the weights given to the spatial gradient constraint in the inversion relative to the straight flux constraint cases ought to be given. Perhaps the spatial gradient case does a better job because it has a looser (or tighter) weighting than the other cases. A tighter flux prior usually results in a worse fit to the measurement data; or, vice versa, the inversion can over-fit the measurement data at the cost of too great a change from the flux prior. Knowing the weights assumed in the inversion for the gradient case vis a vis the straight flux case could help assess this. Similarly, some information on how good the fit to the measurement data is for the four cases could help.

Reply: Thanks for the comments. We fully agree with the reviewer that the weight of a priori in different experiments should be identical. Since we assume that the uncertainty of the two a priori (flux and its spatial gradient) is 5 time larger than than the analysis uncertainty (Text A1, Line 308). To further illustrate the weight of the a priori, figure R1 shows the uncertainty reduced by the $CO_2$ observation and the a priori in different experiments, which varified that the weight of the two a priori are identical in EXP-GOI (spatial gradient) and EXP-FOI (flux). The ensemble uncertainty reduciton (EUR) is defined as,

$$EUR_{i,t} = \frac{\sigma^b_{i,t} - \sigma^a_{i,t}}{\sigma^b_{i,t}},$$

where $\sigma^b_{i,t}$ and $\sigma^a_{i,t}$ is the first guess and final analysis ensemble uncertainty, respectively, at a given grid point i and a given time t. Since there are two types of observation, the EUR can be separated to two parts of $EUR^{CO2}$ and $EUR^{Ap}$ as,

$$EUR^{CO2}_{i,t} = \frac{\sigma^b_{i,t} - \sigma^{a*}_{i,t}}{\sigma^b_{i,t}},$$

$$EUR^{Ap}_{i,t} = \frac{\sigma^{a*}_{i,t} - \sigma^a_{i,t}}{\sigma^b_{i,t}},$$

where the superscript $a*$ denote the analysis after assimilating $CO_2$ observation; the superscript Ap denotes the a priori.

As show in Figure R1, for EXP-GOI, the $EUR^{CO2}$ in the northern middle and high latitudes of North America and Europe, where the $CO_2$ observation network is dense, can exceed 30%. Generally, the $EUR^{CO2}$ decreases from north to south. In South America and Africa, the $EUR^{CO2}$ is around 10%. Since the uncertainty of a priori is set to be proportional to the ensemble uncertainty, ***$EUR^{Ap}$ is approximately 5% and almost identical at different grids. For EXP-FOI, the $EUR^{Ap}$ is configured to be identical to EXP-GOI; thus, the weights of a priori in EXP-GOI and EXP-FOI are expected to be identical.*** However, in EXP-GOI, the $EUR^{Ap}$ in the Qinghai-Tibet Plateau, Andes, Sahel, and Arabian Plateau is relatively larger than in the other areas, suggesting that, even though the error of the gradient at different grid points is identical, the $EUR^{Ap}$ is also affected by the geography. In EXP-GI, which assimilated only

the in-situ data, the $EUR^{CO2}$ is approximately half that in EXP-GOI, resulting in similar $EUR^{CO2}$ and $EUR^{Ap}$ in South America and Africa.

[Figure]

**Figure R1: a-c) The ensemble uncertainty reduction in EXP-GOI from the satellite and in-situ $CO_2$ observations, the a priori of the gradient, and the sum of the former two. d-f) The ensemble uncertainty reduction in EXP-FOI from the satellite and in-situ $CO_2$ observations, the a priori of the original flux, and the sum of the former two. g-i) The ensemble uncertainty reduction in EXP-GOI from the in-situ $CO_2$ observations, the a priori of the gradient, and the sum of the former two.**

Third, if the flux constraint can be implemented equally as well in the measurement vector as in the a priori state vector, then the two cases in which the straight flux prior are implemented these two ways should give the same flux results. That is, the EXP-NP case, in which the flux prior is applied normally, as the a priori constraint on the fluxes in the state vector, and the EXP-AP case, in which the flux prior is assimilated as a measurement in the measurement vector, should give the same flux estimates. But they don't -- they give quite different answers, as seen by the turquois and orange lines in Figures 3 through 5. What is it about the different implementation of the prior that causes these differences? Different weights used in each case? A different number of times that the constraint is applied (if fluxes at multiple times are updated by measurements at a single time)? Similarly in Figures 6 and 7, the EXP-NP case gives much worse RMSEs for flux and flux spatial gradient than does EXP-AP. Why is this, if the two ways of implementing the prior are equivalent? I can understand why, with a short-window inversion, the EXP-NP case might have higher values for these metrics (i.e. a flux error frozen in at a given assimilation step would need to be corrected by a balancing error at the next step of opposite sign, resulting in a lot of noise in time), but what is it about the EXP-AP implementation that prevents this?

Reply: Thanks for the comments. The reviewer may misunderstand the experiment name. The EXP-NP (now EXP-OI) is the experiment that does not use a priori. The much worse RMSEs in EXP-NP is because of lacking regularization. The cases that using the a priori flux in the measurement vector is EXP-P. And comparing between EXP-AP and EXP-P, there differences are small.

The case using the a priori flux in the measurement vector has several differences as compared to the

case using the a priori flux in the a priori vector. Since the AAPO method treat the a priori as a special observation, the a priori can reduce the ensemble uncertainty as show in Figure R1, thus will change the values of each ensemble member. However, when placed in the a priori vector, only the ensemble mean is changed, and the ensemble uncertainty are not reduced.

Furthermore, the two cases are not comparable in terms of weight. The measurement vector case applied observation localization with a small radiu, thus the surrounding a priori information will also influence the local flux estimates. Thus, it is hard for us to make the two cases identical in terms of weight. Considering this point, we decide to delete the case that using the a priori flux in the measurement vector.

Fourth, because the OSSE experiments use the same ocean fluxes in the truth and assimilation runs, there is effectively no error coming from the oceans and no need to allocate any flux corrections there in the inversions. This is effectively the same thing as holding the oceans fixed and only allowing flux changes over the land areas. This significantly simplifies the inversion and gives an overlyoptimistic view of how well the inversions can retrieve the land fluxes. However, even worse, it may favor the spatial gradient prior constraint more than the straight flux prior constraint, since, with the ocean corrections fixed to zero, the fluxes bordering the oceans are then strongly constrained by the spatial gradient constraint, and the fluxes in the interior similarly prevented from moving as much as they otherwise would. With the straight flux constraint, however, the fluxes are still allowed to trade off corrections between continents. It would be interesting to see whether these same favorable results with the EXP-ASG case are achieved if more realistic errors are allowed over the oceans (i.e., if separate ocean flux models were used in generating the truth and prior, as has been done with the land biospheric fluxes here).

Reply: Thanks for the comments and suggestions. We agree with the reviewer that the ocean flux plays an important role in regulating the global carbon cycle, and considering the errors from the ocean will make the OSSEs more realistic. But for an OSSE that designed for validating the AAPO method, the main conclusion that the spatial gradient is better than the flux itself holds when focusing on the land. And some previous OSSE studies do not consider the error from the ocean (e.g., Liu et al., 2014).

Moreover, we did not consider the error from ocean because of the short window feature of COLA. Current bottom-up ocean carbon flux estimates usually report only monthly mean value which does not fit well with the additive inflation step in COLA. The additive inflation requires daily bottom-up estimates. And in real data assimilation experiments, COLA uses a daily ocean carbon flux estimates from Jena Carboscope (Rödenbeck et al., 2014) as the a priori ocean flux. And, to our best knowledge, we do not know other public-available daily ocean flux product (except the CMIP output), which is not practical for us to use another independent daily ocean flux product as the a priori.

As the reviewer point out that the ocean flux may favor the spatial gradient of flux. We are investigating it in the real data assimilation experiments that using the Jena Carboscope estimates as the a priori ocean flux.

*Reference:*

*Rödenbeck, Christian, et al. "Interannual sea–air CO2 flux variability from an observation-driven ocean mixed-layer scheme." Biogeosciences 11.17 (2014): 4599-4613.*

*Liu, Junjie, et al. "Carbon monitoring system flux estimation and attribution: impact of ACOS-GOSAT XCO2 sampling on the inference of terrestrial biospheric sources and sinks." Tellus B: Chemical and Physical Meteorology 66.1 (2014): 22486.*

Fifth, it would be useful for the authors to discuss how specific their results are to the flux inversion method they use (a short-window EnKF). Would they anticipate that the alternative flux spatial gradient constraint would give similar improvements in methods that allow the transport model to link

measurements and flux corrections across a longer span? Similarly, since this reliance on the transport model is less important when there is more data coverage, would the results obtained here still hold were a less-dense observing network (the in situ CO2 network instead of a CO2-measuring satellite, say) to be used?

Reply: Thanks for the suggestions. To answer this question, we conducted extra two experiments (EXP-GI and EXP-FI) that assimilates only the insitu observations.

First, we acknowledge that the short-window based COLA system may not be better than long window-based systems while assimilate only the insitu observations (Line 247): *Since the observation window in COLA is relatively shorter than some traditional systems (e.g., CarbonTracker, UoE, and CAMS), we speculate that long window-based systems should be more suitable for inversions using only surface $CO_2$ observations and constrained by the a priori $F_{TA}$.*

However, comparing EXP-GI and EXP-FI, the main conclusions still hold. The hemispheric partitioning estimates in EXP-GI is less biased even only constrained by the insitu observations. And the seasonal RMSEs in EXP-GI are also smaller.

More-detailed comments:

14: "dynamic constraints" I do not believe that the reason the inversion problem is ill-posed is because of the lack of explicit dynamical constraints in the setup. Really it is due to the sparse data.

Reply: Thanks for the comments. We have deleted the "dynamic constraints" in the abstract.

16-17: "Ensemble Kalman filter-based inversion algorithms usually weigh a priori flux to the background or directly replace the background with the a priori flux." It is not very clear what this means. Please reword. What do you mean by 'background'?

Reply: Thanks for the comments. The 'background' corresponds to the first guess. In this context maybe misleading and reader may refer to the 'background' $CO_2$ concentration. Thus, we replace the 'background' to 'first guess'.

21: spell out "AAPO"? It is not clear why you use this combination of letters for what you are describing.

Reply: Thanks for the comment. We have spell out "AAPO" in the abstract in Line 24, "Assimilates A Priori information as a special Observation (AAPO)".

38: I wouldn't say the problem is 'ill-posed' because of transport errors or retrieval biases -- those just bias the result. Ill-posedness is more due to lack of a sufficient data constraint, for example, trying to solve for more unknowns than can be constrained by a given number of data points.

Reply: Thanks for the comment. We agree that the systematic errors are not the reason of being "ill-posed" but other influence factors. The sentences are revised to be clearer in Line 42, "*However, the top-down estimation could be ill-posed because of the sparseness feature of atmospheric $CO_2$ observations. And systematic errors in the transport model and satellite retrieval can contaminate the illustration of inferred SCFs (Basu et al., 2018; O'Dell et al., 2018; Yu et al., 2018; Schuh et al., 2019)*".

49: "the LETKF with a short assimilation window and long observation window setting" I do not see this described later in the text. Please describe what these 'window' terms refer to, for example in terms of the filter time stepping, what span of data is assimilated at each time step, and what span of fluxes is allowed to change per time step; preferably with equations.

Reply: Thanks for the comment on the windows. We have added more details on describing the windows from Line 88 to Line 95. The end of assimilation window means when update the state and parameter. Within the observation window, the model will forecast the modeled observations in order to match with the 7 days of observations.

54-56: "On the other hand, even though a priori information includes biases, it could be used to further improve the SCF estimation in COLA because it includes important dynamic information generated by terrestrial models, which is missing in the top-down inversion system." It is not clear why you think that dynamic information generated by the terrestrial models is not represented in the top-down inversion systems. Insofar as it is used to generate the a priori SCFs, it is in there. Do you mean to say that the dynamical constraint of the a priori fluxes is not represented explicitly as a dynamic model in the Kalman filter, i.e. as a formal constraint?

Reply: Thanks for the comment and pointing out the mistake. Yes, we intend to say that the ensemble-based COLA system does not hold a dynamic model and a priori can partly compensate the dynamic information.

We revised this sentence to be more accurate in Line 60, " *On the other hand, since the ensemble-based COLA system does not hold a dynamic flux model, a priori fluxes generated by terrestrial models has the potential to further enhance the SCFs estimation in the COLA system*".

75: add "at" after "including"

Reply: Thanks for pointing out the mistake. To make this sentence clearer, we have revised this sentence, "*LETKF is a deterministic variation of EnKF and is known for its efficiency in DA.*".

77-81" Similar to the other EnKF, the LETKF prefers a short assimilation window to produce accurate model state analysis, which reduces noise within the background for parameter estimation. On the other hand, parameter estimation requires a long training period to enhance the model response to the estimated parameter (the signal). Therefore, COLA implements a new version of LETKF with a unique feature of a short assimilation window (1 day) and a long observation window (7 days) to enhance the SCF estimation (Liu et al., 2019)." It is not clear how these various 'windows' relate to the fluxes being solved for. You should write out with equations what is being solved for, how the time stepping is done, what observations are assimilated in which time step with which weights, etc. And point out which spans are the 'observation window' versus the 'assimilation window'. This may be detailed in previous LETKF papers, but the reader shouldn't have to go back to them to understand what is being used here.

Reply: Thanks for the comment on the windows. We have added more details on describing the windows from Line 88 to Line 95. The end of assimilation window means when update the state and parameter. Within the observation window, the model will forecast the modeled observations in order to match with the 7 days of observations. We intend to add an equation to illustrate it, but we find it may add more questions to the readers. The LETKF analysis equation is,

$$\bar{\mathbf{x}}_{t_1}^a = \bar{\mathbf{x}}_{t_1}^b + \mathbf{X}_{t_1}^b \widetilde{\mathbf{P}}^a \left(\mathbf{Y}_{t_0 \to t_7}^b\right)^T \mathbf{R}_{t_0 \to t_7}^{-1} \left(\mathbf{y}_{t_0 \to t_7}^o - \bar{\mathbf{y}}_{t_0 \to t_7}^b\right)$$

$$\widetilde{\mathbf{P}}^a = [\left(\mathbf{Y}_{t_0 \to t_7}^b\right)^T \mathbf{R}_{t_0 \to t_7}^{-1} \left(\mathbf{Y}_{t_0 \to t_7}^b\right) + (K-1)\mathbf{I}]^{-1}$$

where the flux parameter $\mathbf{f}$ is augmented to the $CO_2$ state $\mathbf{c}$ that $\mathbf{x} = [\mathbf{c}, \mathbf{f}]^T$; the superscripts a and b denote the analysis and background (first guess), respectively; $\bar{\mathbf{x}}$ and $\mathbf{X}$ are the ensemble mean and ensemble perturbation, respectively; the subscript $t_1$ indicate the end of assimilation window of 1 day; $\mathbf{y}_{t_0 \to t_7}^o$ is the $CO_2$ observations within the observation window of 7 days; $\mathbf{y}_{t_0 \to t_7}^b$ is the forecasted observations corresponding to each observations; $\mathbf{Y}^b$ is the ensemble perturbation in the observation space; $\mathbf{R}$ is the observation error matrix; $\widetilde{\mathbf{P}}^a$ is the analysis error covariance; K is the ensemble size which is set to 20; and $\mathbf{I}$ is the identical matrix.

In the LETKF analysis equation, the windows are expressed by the subscripts, readers may misunderstand that there are only 7 timesteps of observations. We have discussed for several time and

decide not put it in the manuscript but describing it directly. The descriptions in Line 88 are, "*In this approach, the persistent flux parameter and dynamic CO₂ state are updated on a daily basis using observations within the long observation window (7 days). To achieve that, for each analysis cycle, GEOS-Chem forecasts for 7 days to generate forecast observations, which are then assimilated with the corresponding observations using LETKF. This optimization process updates the model state (CO₂) and parameters (SCFs) at the end of the assimilation window, serving as the forecast initial conditions and timing for the subsequent analysis cycle*".

119: "In COLA, the main purpose of applying a priori regularization is to introduce the dynamic constraint for SCF estimation." It is not at all clear that you have now introduced a better dynamic constraint by changing from using the prior flux value to using spatial gradients instead. Nothing involving dynamics has been changed by this. All you have succeeded in doing is removing the link to the overall absolute value of the prior flux (the long-term mean). That may indeed have value, but don't confuse it with dynamics. Any dynamics that were or were not in the original flux prior are still there with this new constraint. Please reword to reflect this, here and elsewhere in the document where 'dynamics' are discussed.

Reply: Thanks for the comment. In the last version, we intended to use 'dynamic/dynamically' to describe that the added a priori information may partly compensate the loss of a priori fluxes in COLA. The a priori fluxes are generated using dynamic vegetation model, thus the a priori fluxes themselves contains dynamic information. In this context, we acknowledge that the description is very misleading. We have deleted the dynamic/dynamically in some places (e.g., the Title) and rewrite some sentences. For example, in Line 137, "*Within COLA, rather than using the SCFs estimation itself as the a priori information, we propose the utilization of the spatial gradient of a bottom-up estimation of SCFs ($\nabla f$) as a more suitable alternative*".

138-147: You are free to add dynamical noise to your propagation of information forward in time in your model. You should discuss why you choose not to add dynamical noise that reflects errors in your transport model and/or variability in the land fluxes not captured by a forward propagation based on persistence. Why do you instead add an inflation term that is based more on the technical needs of your EnKF rather than a physically-based dynamical error?

Reply: Thanks for the comment. First, in the perfect model OSSEs, we did not consider the transport model error. We acknowlege that there is transport model error while conducting real data assimilation. If you want to consider or reduce the transport error, the error should be related to the transport model by perturbing the meteorology fields instead of the adding noises to the flux ensembles. The transport model error is always a tough question for CO₂ inversion. Several recent studies have discussed the impact of transport model error/differences on modeling the CO₂ concentration (Schuh et al., 2019, 2023). But those studies are performed in forward simulation and considering/reducing transport error in inversions needs to couple with an online general circulation model instead of an offline transport model (Kang et al., 2012).

Second, as the reviewer pointed out, the forward propagation based on the persistence can not capture the noise/uncertainty in the land fluxes. The additive inflation method is designed to add the uncertainty related to the land fluxes physically. We have added the details of the inflation method to the appendix section (Text A1). Based on the inflation method, the noises/variance are added based on the temporal changes of the a priori fluxes.

*Reference:*

*Kang, Ji-Sun, et al. "Estimation of surface carbon fluxes with an advanced data assimilation*

*methodology." Journal of Geophysical Research: Atmospheres 117.D24 (2012).*

149-150: "COLA assimilates the a priori SCF spatial gradients into the system, which needs to define the a priori uncertainty. In this study, we simply set the a priori uncertainty proportional to the uncertainty of the analysis ensemble uncertainty." Please describe what this analysis ensemble uncertainty looks like. Does it differentiate between forested areas that are likely to have larger fluxes and flux uncertainty and desert areas that are likely to have smaller ones? (Or similarly for flux gradients?) A sensitivity study done using uncertainties proportional to the magnitude of the fluxes in either the VEGAS or CASA models, or based on the difference between VEGAS and CASA (and preferably other models), would be welcome to test the dependence of your results on this assumption.

Reply: Thanks for the comment. Yes, the analysis uncertainty of flux or its spatial gradient is large in the northern forest areas and small in the desert. The magnitude of the analysis uncertainty is mainly dependent on the spatial coverage of $CO_2$ observation (Fig. R1) and the additive inflation method described in the appendix (Text A1). In area with more $CO_2$ observations and smaller monthly variation of fluxes, the analysis uncertainty would be smaller. With the details on the additive inflation method, we hope that your questions on the analysis ensemble uncertainty are addressed.

The sensitivity test suggested by the reviewer is interesting. We plan to test these configurations in the future. We discussed the choose of a priori uncertainty in Text A1 (Line 350), "*In reality, a bottom-up SCFs estimation product may come with its uncertainty estimation. We may derive the uncertainty of the SCFs spatial gradient from it. The importance and impact of those uncertainties and whether their accuracies are good enough for DA application remain to be further explored in the future*".

165-166: "We set the CO2 observation localization radius to 4000 kilometers." Since the general reader probably will not understand what this means, please say what this means, practically, in your inversion setup. Does it mean literally that each observation has zero impact on any flux farther away than 4000 kilometers at a given time? What about at previous times?

Reply: Thanks for the suggestion. Miyoshi et al. (2007) described the localization weight w(r) in LETKF as,

$$w(r) = e^{(\frac{r_h^2}{2d_h^2} + \frac{r_v^2}{2d_v^2})},$$

where $d_h$ and $d_v$ denote the horizontal and vertical localization radius; and $r_h$ and $r_v$ is the distance between an observation and a model grid horizontally and vertically. If the distance exceeds $2\sqrt{\frac{10}{3}}d$, the observation will be discarded. And we did not apply temporal localization.

We discussed and decided to put the details to the manuscript and explicit guiding reader to infer Miyoshi et al. (2007) for the information of localization scheme in Line 163.

168-174: By using the same fossil fuel, ocean, and wildfire fluxes in both the truth and prior, the simulation is artificially rosy: terrestrial fluxes are solved for using only differences there by permitting flux corrections only over the land and not over the ocean. By not considering the impact of ocean flux errors, this will give you lower error estimates for the land fluxes than you'd get otherwise. It would be a useful sensitivity study to look at the impact of considering ocean flux errors, as well. Figures 6 & 7: The difference between the EXP-NP and EXP-AP cases still needs to be explained. Yes, the short window of the COLA setup results in over-fitting of the data and noisy fluxes (and spatial gradients) in the EXP-NP case. But how does applying the prior flux constraint via the measurement vector prevent this?

Reply: Thanks for the comments. We agree with the reviewer that the ocean flux plays an important role in regulating the global carbon cycle and considering the errors from the ocean will make the OSSEs

more realistic. But for an OSSE that designed for validating the AAPO method, the main conclusion that the spatial gradient is better than the flux itself holds when focusing on the land. And some previous OSSE studies do not consider the error from the ocean (e.g., Liu et al., 2014). Moreover, we did not consider the error from ocean because of the short window feature of COLA. Current bottom-up ocean carbon flux estimates usually report only monthly mean value which does not fit well with the additive inflation step in COLA. The additive inflation requires daily bottom-up estimates. And in real data assimilation experiments, COLA uses a daily ocean carbon flux estimates from Jena Carboscope (Rödenbeck et al., 2014) as the a priori ocean flux. And, to our best knowledge, we do not know other public-available daily ocean flux product (except the CMIP output), which is not practical for us to use another independent daily ocean flux product as the a priori. As the reviewer point out that the ocean flux may favor the spatial gradient of flux. We are investigating it in the real data assimilation experiments that using the Jena Carboscope estimates as the a priori ocean flux.

The reviewer may misunderstand the experiment name. The EXP-NP (now EXP-OI) is the experiment that does not use a priori. The much worse RMSEs in EXP-NP is because of lacking regularization. The cases that using the a priori flux in the measurement vector is EXP-P. And comparing between EXP-AP and EXP-P, there differences are small. The case using the a priori flux in the measurement vector has several differences as compared to the case using the a priori flux in the a priori vector. Since the AAPO method treat the a priori as a special observation, the a priori can reduce the ensemble uncertainty as show in Figure R1, thus will change the values of each ensemble member. However, when placed in the a priori vector, only the ensemble mean is changed, and the ensemble uncertainty are not reduced. Furthermore, the two cases are not comparable in terms of weight. The measurement vector case applied observation localization with a small radiu, thus the surrounding a priori information will also influence the local flux estimates. Thus, it is hard for us to make the two cases identical in terms of weight. Considering this point, we decide to delete the case that using the a priori flux in the measurement vector.

*Reference:*

*Rödenbeck, Christian, et al. "Interannual sea–air CO2 flux variability from an observation-driven ocean mixed-layer scheme." Biogeosciences 11.17 (2014): 4599-4613.*

*Liu, Junjie, et al. "Carbon monitoring system flux estimation and attribution: impact of ACOS-GOSAT XCO2 sampling on the inference of terrestrial biospheric sources and sinks." Tellus B: Chemical and Physical Meteorology 66.1 (2014): 22486.*

290: What does 'dynamically' in 'dynamically assimilated' indicate? Is this some special sort of assimilation method? Also, define what the acronym 'AAPO' refers to.

Reply: Thanks for the comment and pointing out the mistake. As the reviewer mentioned in the previous comments, the 'dynamically' may mislead readers. And we deleted it in Line 290. And 'AAPO' is defined in the method section. And we revised the sentense in Line 290, "*In this study, we developed a novel algorithm for the ensemble-based COLA $CO_2$ inversion system, in which the spatial gradient of a bottom-up model estimation is assimilated as a special observation*".

297-304: "However, the advantage of error transport is partly sacrificed or abandoned by introducing the a priori flux information to the background in most of the EnKF-based CO2 inversion methods (Peters et al., 2007; Feng et al., 2009). This is because of the loss of a dynamic model to provide the background and the background covariance estimations. Different from most EnKF-based systems, COLA maintains the mean and error transport advantages of the EnKF by including the dynamic information constraints of the a priori flux spatial gradient and using an additive covariance inflation method (Liu et al., 2022)."
I agree that the loss of the dynamical model for the fluxes in most of our flux inversion methodologies

is unfortunate. I do not believe, however, that you are remedying that with your spatial gradient constraint here. Nothing has changed regarding the dynamics in using this constraint. Your only change is to cut the tie to the long-term mean, allowing your estimate to be shifted up or down as a whole more easily.

Reply: Thanks for the comments. We acknowledge that this description is exaggerated. We deleted this paragraph to avoid misunderstanding.

310: 'unique strategy'? Maybe referring to it as a 'new strategy' would be better.

Reply: Thanks for the comments. We have revised to "novel strategy".

**Table 1: Assimilation experiments setup.**

| Experiment | EXP-GOI | EXP-FOI | EXP-OI | EXP-GI | EXP-FI |
|---|---|---|---|---|---|
| A priori | $\nabla f$ | $f$ | | $\nabla f$ | $f$ |
| Observation | OCO-2+In-situ | OCO-2+In-situ | OCO-2+In-situ | In-situ | In-situ |

[Figure]

**Figure 1: The assimilation cycle of the COLA system, illustrating how and where the a priori is assimilated.**

[Figure]

**Figure 2: a) The annual mean signal-to-noise ratio pattern of the a priori $F_{TA}$ spatial gradient. b) The same as a) but for the a priori $F_{TA}$. c) Annual mean signal-to-noise ratio of the spatial gradient divided by the signal-to-noise ratio of the $F_{TA}$.**

[Figure]

**Figure 3: The annual mean F_TA in 2015 of a) the truth, b) the a priori, c) EXP-GOI, d) EXP-FOI, e) EXP-GI, and f) EXP-FI. g) Comparison of the annual total F_TA in the northern extratropical area and the tropical and southern extratropical areas. Different scatters denote the truth, the a priori, and the different experiments. The dashed black line denotes the global carbon budget.**

[Figure]

**Figure 4: The top figures in each subplot are the a) global, b) northern extratropical, and c) tropical and southern extratropical seasonal cycles of F_TA in the truth (black), the a priori (gray), EXP-GOI (red), EXP-FOI (orange), and EXP-OI (sky blue). The bottom figures in each subplot are the global total difference of the a priori and the three assimilation experiments compared to the truth. The annual mean RMSEs of the a priori F_TA and the three experiments are denoted at the upper right corner of the bottom figures.**

[Figure]

**Figure 5: The regional RMSE of the a priori and the five assimilation experiments compared to the truth. The regions are defined by the OCO2MIP.**